# WISE: World Knowledge-Informed Semantic Evaluation for Text-to-Image Generation

**Yuwei Niu** [* 1 2 3]  **Munan Ning** [* 1]  **Mengren Zheng** [3]  **Weiyang Jin** [4]  **Bin Lin** [1]  **Peng Jin** [1]  **Jiaqi Liao** [1]
**Chaoran Feng** [1]  **Fanqing Meng** [5]  **Kunpeng Ning** [1]  **Bin Zhu** [1]  **Li Yuan** [1 2]

## Abstract

Text-to-Image (T2I) models are capable of generating high-quality artistic creations and visual content. However, existing research and evaluation standards predominantly focus on image realism and shallow text-image alignment, leaving complex semantic understanding and world knowledge integration insufficiently evaluated. To address this challenge, we propose **WISE**, a benchmark specifically designed for **W**orld Knowledge-**I**nformed **S**emantic **E**valuation. WISE moves beyond simple word-pixel mapping by challenging models with 1,000 carefully curated prompts across 25 subdomains in cultural common sense, spatio-temporal reasoning, and natural science. We evaluate 20 models (10 dedicated T2I models and 10 unified multimodal models) on these prompts. The results reveal clear limitations in models' ability to integrate and apply world knowledge during image generation, pointing to the need for stronger knowledge incorporation in next-generation T2I models. Code and data are available at https://github.com/PKU-YuanGroup/WISE.

## 1. Introduction

Text-to-image (T2I) models (Zhang et al., 2023) can generate high-quality images that visually match explicit text descriptions. However, they often struggle to ensure factual accuracy, especially for prompts that require complex semantic understanding and world knowledge. This shortcoming is closely related to their limited ability to represent and use world knowledge (Yu et al., 2023), including the broad

facts and complex relationships required for real-world understanding. Although current unified multimodal models have begun to use the powerful text modeling and implicit information extraction capabilities of LLMs to address this bottleneck through specialized representations and generative decoders (Pan et al., 2025; Tong et al., 2024), existing evaluation benchmarks remain limited. They mainly focus on surface-level image-text alignment and fail to evaluate the core capabilities of T2I models from the perspective of implicit understanding and intrinsic knowledge matching, which hinders the development of intelligent T2I systems in knowledge-intensive scenarios.

Specifically, most T2I benchmarks suffer from a lack of semantic complexity. As shown in Figure 1, they use overly straightforward and simple prompts, failing to effectively challenge models' ability to understand and generate images based on the model's world knowledge.

Furthermore, the most commonly used metric, FID (Heusel et al., 2017), primarily focuses on the realism of generated images. Some benchmarks (Hessel et al., 2021; Wu et al., 2023; Xu et al., 2024) use models such as CLIP (Radford et al., 2021) to assess image-text semantic consistency. However, CLIP's limitations (Yuksekgonul et al., 2022) in capturing fine-grained semantic information and handling complex reasoning make it difficult to evaluate models' ability to process intricate semantic information. Consequently, existing evaluations fail to fully reveal model capabilities in real-world scenarios, particularly in tasks requiring world knowledge. For instance, when generating an image depicting a "tadpole that has undergone metamorphosis," a model needs not only to comprehend the textual description ("tadpole", "metamorphosis") but also to invoke its internal world knowledge. This includes understanding amphibian development, the specific morphological changes involved (e.g., the growth of legs, the loss of the tail, the development of lungs), and the biological processes driving this transformation.

This evaluation gap is especially important with the emergence of unified multimodal models. Since these models are built upon large multimodal or language backbones, they are expected to possess richer world knowledge and

---

[*]Equal contribution  [1]Shenzhen Graduate School, Peking University  [2]PengCheng Laboratory  [3]Chongqing University  [4]The University of Hong Kong  [5]National University of Singapore. Correspondence to: Li Yuan <yuanli-ece@pku.edu.cn>.

*Proceedings of the $43^{rd}$ International Conference on Machine Learning*, Seoul, South Korea. PMLR 306, 2026. Copyright 2026 by the author(s).

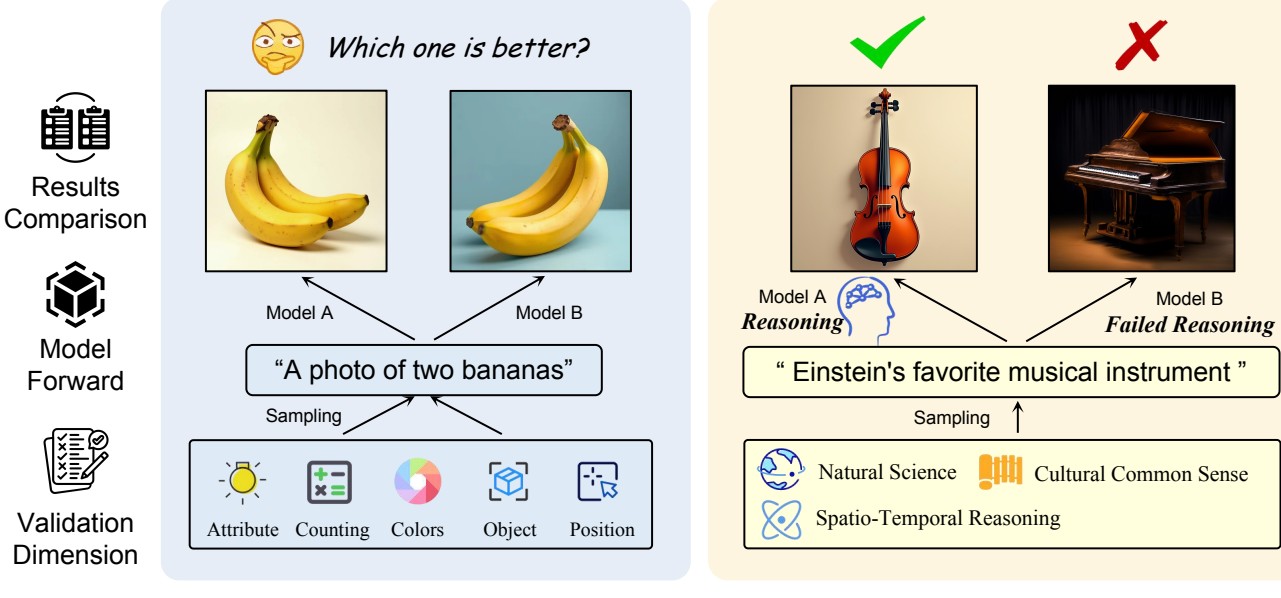

*Figure 1.* Comparison of previous straightforward benchmarks and our proposed WISE. (a) Previous benchmarks typically use simple prompts, such as "A photo of two bananas" in GenEval (Ghosh et al., 2024), which only require shallow text-image alignment. (b) WISE, in contrast, uses prompts that demand world knowledge and reasoning, such as "Einstein's favorite musical instrument," to evaluate a model's ability to generate images based on deeper understanding.

stronger reasoning abilities than conventional T2I systems. However, existing benchmarks provide limited evidence on whether such understanding capabilities actually translate into knowledge-grounded image generation. While some studies have begun to explore whether the strong understanding capabilities of these unified models can benefit image generation, they often rely on overly simplistic benchmarks and therefore provide limited evidence for this phenomenon.

To address this problem, we propose a new benchmark **WISE** (**W**orld Knowledge-**I**nformed **S**emantic **E**valuation). This benchmark systematically evaluates the implicit semantic understanding and world knowledge integration capabilities of T2I models beyond basic text-image alignment through indirect textual cues. WISE covers three major areas: natural sciences, spatiotemporal reasoning, and cultural common sense, and contains 1,000 evaluation questions in 25 sub-areas. For evaluation, we use **WiScore** as a task-specific companion protocol that weights consistency, realism, and aesthetic quality, with consistency emphasized to reflect WISE's focus on knowledge-grounded semantic correctness.

We broadened the scope of our evaluation beyond traditional dedicated T2I models. We use WISE to evaluate a total of 20 T2I models, encompassing both 10 dedicated T2I models and 10 unified multimodal models. However, experiment results demonstrate clear deficiencies in complex semantic understanding and world knowledge integration across

existing T2I models. Even for unified multimodal models, their strong understanding capabilities do not fully translate into advantages in image generation, as revealed by our WISE evaluation. This indicates that current approaches to integrating LLMs within unified multimodal models may not yet fully unlock their potential for image generation that integrates and applies world knowledge.

Our main contributions are as follows:

- We introduce **WISE**, which is designed to evaluate the world knowledge representation capabilities of T2I models through 1,000 carefully curated prompts covering cultural common sense, spatiotemporal reasoning, and natural sciences across 25 sub-areas, rather than traditional simplistic prompts.

- Our experiment results highlight limitations in current T2I models' ability to integrate and apply world knowledge during image generation, revealing directions for future model optimization.

## 2. Related Works

Text-to-image (T2I) generation models, which aim to generate high-quality and diverse images from text, have garnered significant attention recently. These models fall into two main categories: Dedicated T2I Models and Unified Multimodal Models. We review both lines of work before discussing evaluation protocols.

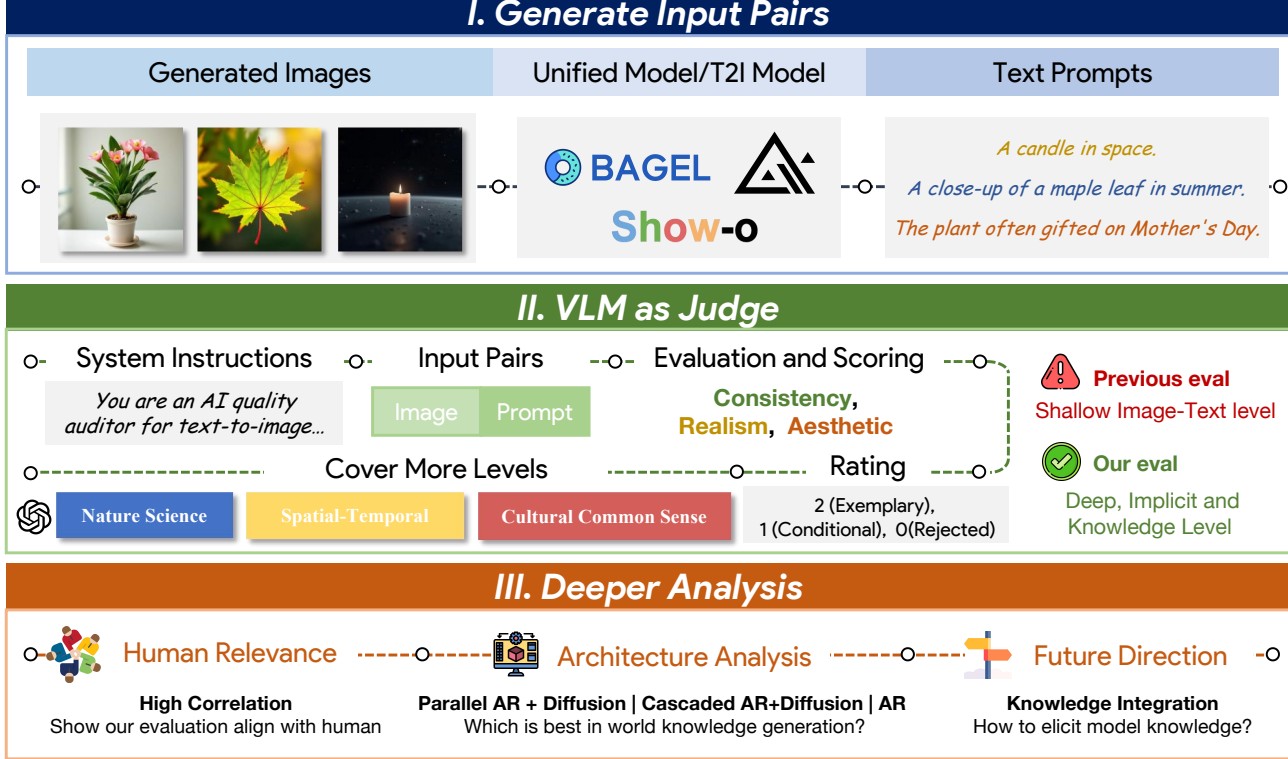

*Figure 2.* Illustration of the WISE framework, which employs a three-stage evaluation process (Panel I to III) to systematically evaluate generated content across three core dimensions. In the two representative cases, the science-domain input "candle in space" violates oxygen-dependent combustion principles, while the spatiotemporal-domain input "close-up of summer maple leaf" contradicts botanical seasonal patterns. Both receive 0 in consistency (see Evaluation Metrics in Panel II), confirming the benchmark's sensitivity to world-knowledge conflicts.

## 2.1. Dedicated T2I Models

Dedicated T2I models represent the mainstream approach in the T2I field and have made rapid progress in recent research. Currently, these models primarily fall into two categories: autoregressive models and diffusion models. Autoregressive (Chen et al., 2020; Fan et al., 2024; Han et al., 2025; Tian et al., 2024; Sun et al., 2024a) models treat image generation as a sequence generation problem, similar to text generation. However, due to their computational cost and limitations in image quality, diffusion models have become the dominant paradigm. Diffusion (Ho et al., 2020; Ramesh et al., 2022; Labs, 2024; Saharia et al., 2022) models iteratively add noise to an image and then progressively denoise it, often using pre-trained text encoders (e.g., CLIP (Radford et al., 2021)) to transform text prompts into embeddings that guide the denoising process. Key advancements include GLIDE (Nichol et al., 2021), which pioneered diffusion models for T2I; Latent Diffusion Models (LDMs) (Rombach et al., 2022), which improve quality and efficiency by operating in latent space; and Stable Diffusion series (Podell et al., 2023; Esser et al., 2024), a landmark achievement built on LDMs.

## 2.2. Unified Multimodal Models

Unified multimodal models aim to build general-purpose systems that can process both textual and visual inputs and perform cross-modal generation and understanding. These models (Team, 2024; Jin et al., 2023; Ge et al., 2024; Diao et al., 2026; Wu et al., 2024a; Ma et al., 2024; Chen et al., 2025b; Wu et al., 2024d; Kou et al., 2024; Wu et al., 2024b; Li et al., 2024c; Sun et al., 2024b; Tong et al., 2024; Shi et al., 2024; Deng et al., 2025; Lin et al., 2025; Li et al., 2025; Gao et al., 2025; Wu et al., 2025b) are typically built upon powerful large language models (LLMs) (Zhao et al., 2023) and extend next-token prediction (Chen et al., 2024) to image generation: the LLM generates visual tokens, and a VQ-VAE (Van Den Oord et al., 2017) or diffusion model serves as a detokenizer. Moreover, Transfusion (Zhou et al., 2024) and Show-O (Xie et al., 2024) demonstrate that bidirectional image diffusion can be combined with autoregressive text prediction within the same framework. D-DiT (Li et al., 2024d) achieves both Text-to-Image and Image-to-Text tasks using an end-to-end diffusion model. A crucial question for unified multimodal models is whether their understanding and generation capabilities can mutually en-

hance each other. Some studies (Tong et al., 2024; Wu et al., 2024b) have provided evidence supporting this phenomenon. However, compared with comprehensive benchmarks for multimodal understanding, T2I benchmarks are often relatively simple and lack in-depth examination of complex semantic understanding and world knowledge reasoning, making this effect difficult to verify. Consequently, a new benchmark is needed to probe these emerging, knowledge-driven generative capabilities.

## 2.3. Text to Image Evaluation

Despite the Fréchet Inception Distance (FID) (Heusel et al., 2017) being one of the most widely adopted metrics for evaluating the quality of generated images, it falls short in assessing text-image consistency, thus failing to comprehensively measure the capabilities of text-to-image models. To address this deficiency, researchers have introduced a series of more sophisticated and challenging benchmarks (Sim et al., 2024; Lin et al., 2024; Wu et al., 2024c) and evaluation metrics (Hessel et al., 2021; Li et al., 2024a; Lin et al., 2024; Jin et al., 2025). For instance, DPG-Bench (Hu et al., 2024) focuses on evaluating models' ability in dense prompt following. T2I-CompBench (Huang et al., 2023) provides a benchmark suite for evaluating compositional generation, where prompts typically combine multiple explicit attributes. GenEval (Ghosh et al., 2024) further assesses object-centric compositional attributes such as object co-occurrence, position, number, and color. These benchmarks mainly test whether models follow literal instructions in the prompt.

Recent work has begun to study knowledge- or reasoning-intensive generation. ScImage (Zhang et al., 2025b) evaluates scientific image generation with spatial, numerical, and attribute constraints, including code-based outputs such as Python or TikZ. PhyBench (Meng et al., 2024) and Commonsense-T2I (Fu et al., 2024) focus on physical and commonsense knowledge, respectively. MMMG (Luo et al., 2025) evaluates knowledge-image generation across disciplines with knowledge-graph-based scoring, while WorldGenBench (Zhang et al., 2025a) evaluates world-knowledge grounding with checklist-style semantic expectations. In contrast, WISE targets natural-image generation from implicit prompts across cultural common sense, spatio-temporal reasoning, and natural science, emphasizing whether models can infer unstated visual targets and render them in semantically grounded images.

# 3. The World Knowledge-Informed Semantic Evaluation (WISE) Benchmark

Existing text-based image evaluation systems have two limitations: traditional benchmarks lack in-depth exploration of world knowledge, and mainstream metrics for measuring image-text alignment mainly focus on surface-level se-

mantics. We therefore describe how to construct a world knowledge-informed T2I benchmark in subsection 3.1 and explain the need for a robust metric in subsection 3.2.

## 3.1. Building a Benchmark Based on World Knowledge

Most existing benchmarks adopt prompt designs that are overly straightforward and lack semantic complexity. They primarily evaluate whether models can combine visual elements as explicitly instructed, but fail to assess the models' capacity for complex semantic understanding and integration of world knowledge in text-to-image generation. Going beyond simply mapping words to pixels, we propose to evaluate world knowledge of current T2I models, which refers to the vast and diverse information, facts, and relationships that constitute our understanding of the real world. In our work, we focus on common world knowledge that can be represented visually. As shown in Figure 3, WISE comprises 1000 prompts designed to assess T2I models' understanding and application of this knowledge across three major domains: Cultural Common Sense, Spatio-temporal Reasoning, and Natural Science, which are divided into 25 subdomains.

**Data Collection and Prompt Design.** We collected prompts from a variety of sources, including educational materials, encyclopedias, common sense problem sets, and synthetic data generated by LLMs. These initial prompts were then refined and extended by human annotators to ensure their clarity, complexity, and explicit ground truth. Each prompt is accompanied by an explanation that describes the world knowledge and reasoning required for a potentially successful image generation. We summarize the sourcing and quality-control process in Table 3. Specifically, the details of our three parts are as follows:

*Cultural common sense.* The Cultural Common Sense domain of WISE aims to evaluate the ability of models to understand specific cultural knowledge and apply it to image generation, which reflects the key aspects of understanding the real world. Models should not only understand the visual features of objects (e.g., shape, size, and color), but also match them with cultural knowledge of the real world. A model lacking such intrinsic knowledge matching will be an unintelligent image generator that can only grasp the surface correspondence between text nouns and visual elements, but cannot understand the cultural role of objects in the real world. This section covers a wide range of topics and is subdivided into 10 fine-grained sub-domains, including festivals, sports, religion, crafts, architecture, animals, plants, art, celebrities, and daily life. Together, these categories cover a wide range of cultural experiences and knowledge of humans. For example, prompts may involve generating images related to traditional festival customs, characteristic ethnic crafts, iconic landmarks, animals and plants with

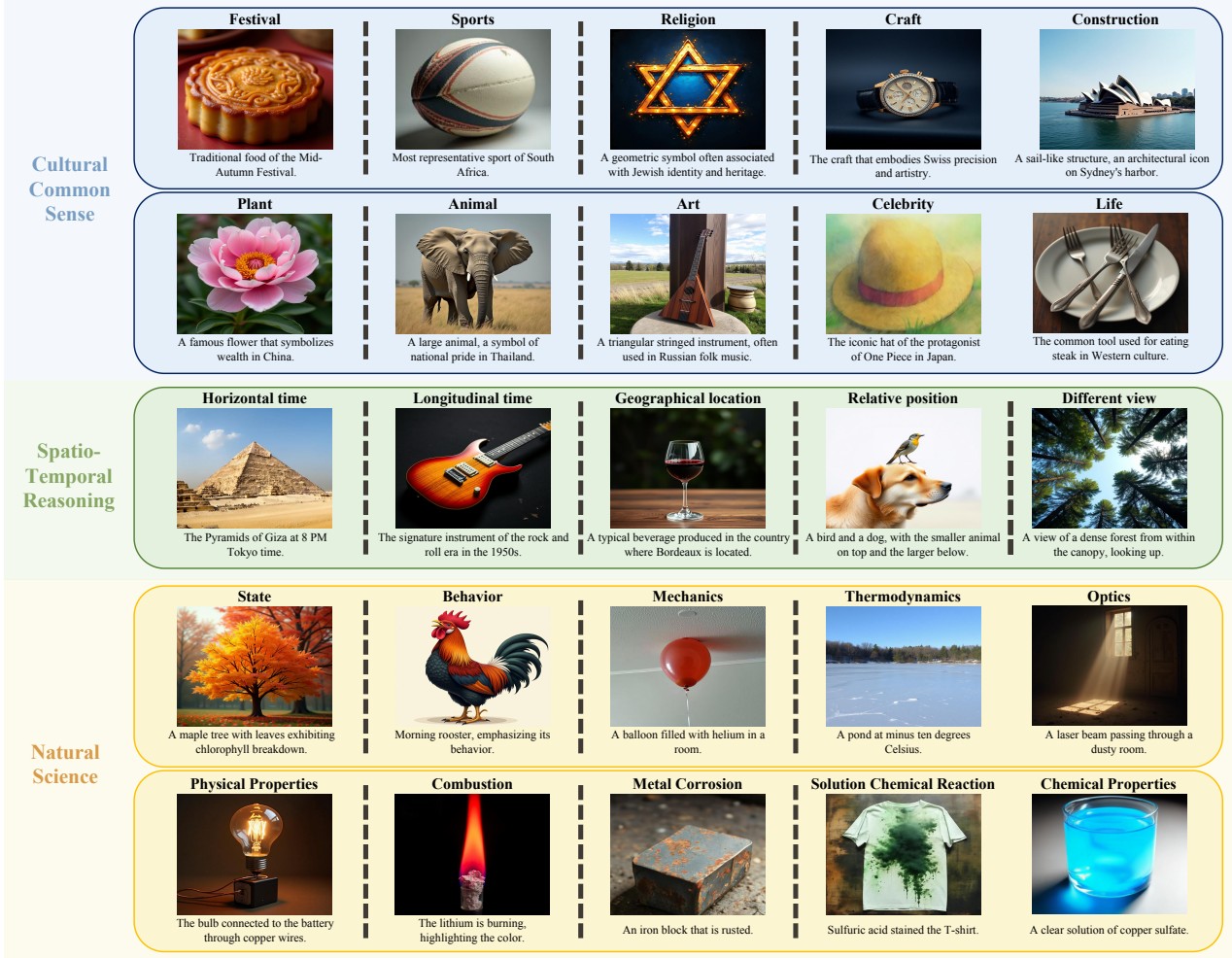

*Figure 3.* Illustrative samples from WISE, covering 3 core dimensions and 25 subdomains. By using non-straightforward semantic prompts, WISE requires T2I models to perform logical inference grounded in world knowledge to accurately generate target entities.

specific cultural significance, common sense in daily life, or events and objects related to celebrities.

*Spatiotemporal reasoning.* The spatiotemporal reasoning domain in WISE is structured around two key dimensions: temporal reasoning and spatial reasoning. Temporal reasoning is divided into Horizontal Temporal reasoning, which assesses understanding of relative temporal relationships between events or objects (e.g., "The Statue of Liberty at 10 pm Dubai time"), and Longitudinal Temporal reasoning, which assesses understanding of absolute temporal relationships, involving specific points in time (e.g., morning, noon, evening, season, specific year, or century). Spatial reasoning is divided into three subcategories: Different Views, which tests understanding of different perspectives, including top view, bottom view, side view, mirror image, and perspective effects; Geographic Relationships, which assesses understanding of spatial relationships between cities, countries, continents, and other geographic entities; and Relative Position, which focuses on understanding the spatial arrangement of objects in a scene relative to each other.

*Natural sciences.* Finally, the WISE benchmark includes a natural sciences domain that aims to assess whether models can not only understand scientific knowledge in a specific domain, but also use this understanding to reason about complex scientific scenarios and generate accurate and scientifically consistent images. At its core, generative models simulate the real world. Our goal is to assess whether these models' understanding goes beyond mere visual replication and encompasses the complex science behind real-world phenomena. For example, a model should not only be able to generate images of water, ice, and steam, but also understand the underlying thermodynamic principles that govern transitions between these states (e.g., freezing, evaporation, condensation). This domain goes beyond general knowledge and delves into specialized bodies of knowledge in biology, physics, and chemistry.

Across all three domains, WISE prompts avoid directly naming the target visual entity whenever possible. Instead, each prompt provides a clue, relation, or condition, requiring models to retrieve world knowledge before generation.

### 3.2. Discovering the Deep Visual Language Alignment Bottlenecks of Traditional Evaluation

Existing metrics such as CLIP-Score (Hessel et al., 2021) and VQA-Score (Li et al., 2024a) offer partial solutions for evaluating text-to-image alignment, but both fall short when faced with complex, knowledge-intensive prompts. CLIP-Score, constrained by CLIP's bag-of-words limitation, struggles with compositional semantics and relational reasoning. VQA-Score improves on this by leveraging likelihood estimation via question answering, but still lacks sensitivity to implicit visual understanding and world knowledge grounding. To address these limitations, we propose a multi-faceted evaluation protocol to rigorously assess the quality of generated images, focusing on four key aspects: Consistency, Realism, Aesthetic Quality, and a composite metric, WiScore.

Consistency evaluates the accuracy and completeness with which the generated image reflects the user's prompt, capturing all key elements and nuances. Realism assesses the realism of the image, considering adherence to physical laws, accurate material representation, and coherent spatial relationships, determining how closely the image resembles a real photograph. Aesthetic Quality measures the overall artistic appeal and visual quality of the image, encompassing aspects such as composition, color harmony, and artistic style. WiScore, the central metric, emphasizes the accuracy of the depicted objects or entities within the generated image, directly reflecting our benchmark's focus on world knowledge utilization. Each component score ranges from 0 to 2, and WiScore normalizes their weighted sum to the range $[0, 1]$:

$$\frac{\alpha_1 \times \text{Consistency} + \alpha_2 \times \text{Realism} + \alpha_3 \times \text{Aesthetic Quality}}{2},$$
$$\text{subject to } \alpha_1 + \alpha_2 + \alpha_3 = 1.$$
$$(1)$$

In this paper, we set $\alpha_1 = 0.7$, $\alpha_2 = 0.2$, and $\alpha_3 = 0.1$. This configuration prioritizes Consistency, reflecting the importance of accurately representing the prompt's intended objects and their relationships, while still incorporating Realism and Aesthetic Quality to ensure overall image quality. A higher WiScore indicates stronger performance in accurately depicting objects and concepts based on world knowledge. The detailed scoring criteria are provided in Appendix D. In the evaluation, we employ GPT-4o-2024-05-13, a powerful multimodal large language model, as the evaluator to assess the performance of T2I models. We adopt a carefully designed scoring protocol to ensure consistency and reliability, with full implementation details provided in Appendix E. The metric comparison further shows that WiScore exhibits the highest agreement with human judgments among all automated metrics. The detailed comparison process is provided in Appendix I.

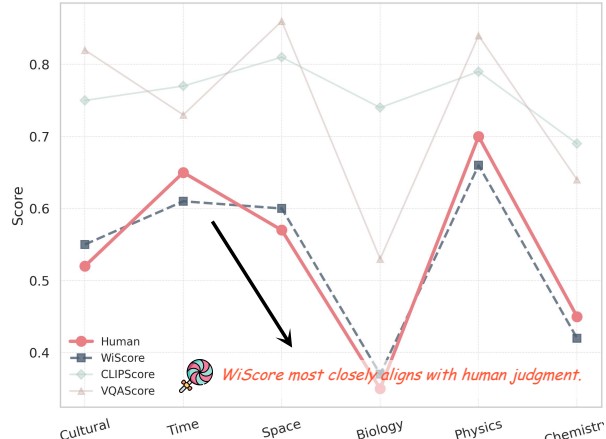

*Figure 4.* Evaluation results of different metrics, where WiScore most closely aligns with human assessment.

## 4. Evaluation Results

### 4.1. Experiment Settings

We evaluate 20 T2I models, including 10 dedicated T2I models and 10 unified multimodal models. The dedicated T2I models are: stable-diffusion-v1-5 (Rombach et al., 2022), stable-diffusion-2-1 (Rombach et al., 2022), stable-diffusion-xl-base-0.9 (Podell et al., 2023), stable-diffusion-3-medium (Esser et al., 2024), stable-diffusion-3.5-medium (Esser et al., 2024), stable-diffusion-3.5-large (Esser et al., 2024), playground-v2.5-1024px-aesthetic (Li et al., 2024b), PixArt-XL-2-1024-MS (Chen et al., 2023), FLUX.1-dev (Labs, 2024), and FLUX.1-schnell (Labs, 2024). For unified multimodal models, we selected 10 models representing three architectural paradigms for evaluation. These include: **Autoregressive (AR)** models, such as Janus-Pro-7B (Chen et al., 2025b), Janus-Pro-1B (Chen et al., 2025b), Janus-1.3B (Wu et al., 2024a), vila-u-7b-256 (Wu et al., 2024d), and Emu3 (Wang et al., 2024), which generate images through next-token prediction over visual tokens; **AR+Diffusion (shallow fusion)** models, where an AR or LLM-style backbone extracts conditional features for a diffusion generator rather than necessarily autoregressively producing visual latent tokens, including Qwen-Image (Wu et al., 2025a), UniWorld (Lin et al., 2025), BLIP3o (Chen et al., 2025a), and MetaQuery (Pan et al., 2025); and the **AR+Diffusion (deep fusion)** model, BAGEL (Deng et al., 2025), which unifies understanding and generation within a deeply integrated transformer framework. For image generation, we use the official default configurations of each model and fix the random seed to ensure reproducibility. For brevity, we use SD to denote stable-diffusion, playground-v2.5 for playground-v2.5-1024px-aesthetic, and PixArt-Alpha for PixArt-XL-2-1024-MS.

*Table 1.* WiScore results across six knowledge categories. Overall denotes the weighted average across categories. Bold indicates the best score within each group.

| Model | Cultural | Time | Space | Biology | Physics | Chemistry | **Overall** |
|---|---|---|---|---|---|---|---|
| **Dedicated T2I** | | | | | | | |
| FLUX.1-dev | 0.48 | **0.58** | **0.62** | 0.42 | 0.51 | **0.35** | **0.50** |
| FLUX.1-schnell | 0.39 | 0.44 | 0.50 | 0.31 | 0.44 | 0.26 | 0.40 |
| PixArt-Alpha | 0.45 | 0.50 | 0.48 | **0.49** | **0.56** | 0.34 | 0.47 |
| playground-v2.5 | **0.49** | 0.58 | 0.55 | 0.43 | 0.48 | 0.33 | 0.49 |
| SD-v1-5 | 0.34 | 0.35 | 0.32 | 0.28 | 0.29 | 0.21 | 0.32 |
| SD-2-1 | 0.30 | 0.38 | 0.35 | 0.33 | 0.34 | 0.21 | 0.32 |
| SD-XL-base-0.9 | 0.43 | 0.48 | 0.47 | 0.44 | 0.45 | 0.27 | 0.43 |
| SD-3-medium | 0.42 | 0.44 | 0.48 | 0.39 | 0.47 | 0.29 | 0.42 |
| SD-3.5-medium | 0.43 | 0.50 | 0.52 | 0.41 | 0.53 | 0.33 | 0.45 |
| SD-3.5-large | 0.44 | 0.50 | 0.58 | 0.44 | 0.52 | 0.31 | 0.46 |
| **Unified MLLM** | | | | | | | |
| *AR+Diffusion (deep fusion)* | | | | | | | |
| BAGEL | 0.44 | 0.55 | 0.68 | 0.44 | 0.60 | 0.39 | 0.52 |
| BAGEL+CoT | **0.76** | **0.69** | 0.75 | **0.65** | **0.75** | **0.58** | **0.70** |
| *AR+Diffusion (shallow fusion)* | | | | | | | |
| Qwen-Image | 0.62 | 0.63 | **0.77** | 0.57 | **0.75** | 0.40 | 0.62 |
| UniWorld-V1 | 0.53 | 0.55 | 0.73 | 0.45 | 0.59 | 0.41 | 0.55 |
| MetaQuery-XL | 0.56 | 0.55 | 0.62 | 0.49 | 0.63 | 0.41 | 0.55 |
| BLIP3o-8B | 0.49 | 0.51 | 0.63 | 0.54 | 0.63 | 0.37 | 0.52 |
| *Autoregressive* | | | | | | | |
| Emu3 | 0.34 | 0.45 | 0.48 | 0.41 | 0.45 | 0.27 | 0.39 |
| Janus-Pro-7B | 0.30 | 0.37 | 0.49 | 0.36 | 0.42 | 0.26 | 0.35 |
| Janus-Pro-1B | 0.20 | 0.28 | 0.45 | 0.24 | 0.32 | 0.16 | 0.26 |
| Janus-1.3B | 0.16 | 0.26 | 0.35 | 0.28 | 0.30 | 0.14 | 0.23 |
| vila-u-7b-256 | 0.26 | 0.33 | 0.37 | 0.35 | 0.39 | 0.23 | 0.31 |

## 4.2. Main Results

**Understanding-Generation Gap in T2I Tasks.** Table 1 presents the WiScore of various models on the WISE benchmark, designed to evaluate the extent to which models leverage world knowledge in image generation. Overall, the majority of models fail to achieve a satisfactory score ($> 0.6$), indicating significant deficiencies in the ability of current models to leverage complex semantic understanding (implicit understanding) and world knowledge (intrinsic knowledge matching) for image generation. This limitation is particularly evident in dedicated T2I models.

At the category level, science-related prompts are generally harder, with Chemistry showing the lowest scores. This is because chemistry prompts often require models to reason about implicit scientific knowledge, such as material properties, reaction processes, solution colors, or corrosion states, and translate these constraints into visual details. These results expose a gap between scientific knowledge retrieval

and faithful visual generation.

Moreover, when examining the overall **model architectures**, we find that the *AR+Diffusion (deep fusion)* and *AR+Diffusion (shallow fusion)* paradigms within unified MLLMs outperform dedicated T2I models, achieving the best results. This suggests that these two architectural approaches are most effective at eliciting the model's internal knowledge, demonstrating the advantages and potential of unified multimodal models.

Notably, when BAGEL uses Chain of Thought (CoT) to assist its visual generation, it achieves the best performance in Table 1. This indicates that CoT is an effective mechanism for eliciting model knowledge, and suggests that training with CoT is a useful strategy for enhancing the application of internal knowledge during generation.

Additional ablations on metric consistency, judge stability, and recent model architectures are provided in Appendices B, C, and J, respectively.

*Table 2.* WiScore on rewritten prompts of different models. These prompts were simplified using GPT-4o (e.g., "The plant often gifted on Mother's Day" to "Carnation"). **Green bold** indicates score increase after rewriting; **red bold** indicates score decrease.

| Model | Cultural | Time | Space | Biology | Physics | Chemistry | **Overall** |
|---|---|---|---|---|---|---|---|
| | | | | **Dedicated T2I** | | | |
| FLUX.1-dev | **0.75** **+0.27** | **0.70** **+0.12** | **0.76** **+0.14** | **0.69** **+0.27** | **0.71** **+0.20** | **0.68** **+0.33** | **0.73** **+0.23** |
| FLUX.1-schnell | 0.63 **+0.24** | 0.58 **+0.14** | 0.67 **+0.17** | 0.58 **+0.27** | 0.58 **+0.14** | 0.44 **+0.18** | 0.60 **+0.20** |
| PixArt-Alpha | 0.66 **+0.21** | 0.64 **+0.14** | 0.55 **+0.07** | 0.58 **+0.09** | 0.64 **+0.08** | 0.62 **+0.28** | 0.63 **+0.16** |
| playground-v2.5 | 0.78 **+0.29** | 0.72 **+0.14** | 0.63 **+0.08** | **0.69** **+0.26** | 0.67 **+0.19** | 0.60 **+0.27** | 0.71 **+0.22** |
| SD-v1-5 | 0.59 **+0.25** | 0.50 **+0.15** | 0.41 **+0.09** | 0.47 **+0.19** | 0.44 **+0.15** | 0.36 **+0.15** | 0.50 **+0.18** |
| SD-2-1 | 0.63 **+0.33** | 0.61 **+0.23** | 0.44 **+0.09** | 0.50 **+0.17** | 0.49 **+0.15** | 0.41 **+0.20** | 0.55 **+0.23** |
| SD-XL-base-0.9 | 0.68 **+0.25** | 0.71 **+0.23** | 0.59 **+0.12** | 0.61 **+0.17** | 0.67 **+0.22** | 0.55 **+0.28** | 0.65 **+0.22** |
| SD-3-medium | 0.76 **+0.34** | 0.65 **+0.21** | 0.68 **+0.20** | 0.59 **+0.20** | 0.67 **+0.20** | 0.59 **+0.30** | 0.69 **+0.27** |
| SD-3.5-medium | 0.73 **+0.30** | 0.69 **+0.19** | 0.67 **+0.15** | 0.68 **+0.27** | 0.67 **+0.14** | 0.60 **+0.27** | 0.69 **+0.24** |
| SD-3.5-large | **0.78** **+0.34** | 0.69 **+0.19** | 0.68 **+0.10** | 0.64 **+0.20** | 0.70 **+0.18** | 0.64 **+0.33** | 0.72 **+0.26** |
| | | | | **Unified MLLM** | | | |
| | | | | *AR+Diffusion (deep fusion)* | | | |
| BAGEL | 0.68 **+0.24** | 0.75 **+0.20** | 0.75 **+0.07** | 0.71 **+0.27** | 0.81 **+0.21** | **0.81** **+0.42** | 0.73 **+0.21** |
| | | | | *AR+Diffusion (shallow fusion)* | | | |
| Qwen-Image | **0.91** **+0.29** | **0.82** **+0.19** | **0.92** **+0.15** | **0.88** **+0.31** | **0.91** **+0.16** | 0.80 **+0.40** | **0.88** **+0.26** |
| UniWorld-V1 | 0.81 **+0.28** | 0.75 **+0.20** | 0.84 **+0.11** | 0.71 **+0.26** | 0.76 **+0.17** | 0.69 **+0.28** | 0.78 **+0.23** |
| MetaQuery-XL | 0.81 **+0.25** | 0.73 **+0.18** | 0.69 **+0.07** | 0.67 **+0.18** | 0.59 **-0.04** | 0.58 **+0.17** | 0.72 **+0.17** |
| BLIP3o-8B | 0.80 **+0.31** | 0.70 **+0.19** | 0.75 **+0.12** | 0.80 **+0.26** | 0.76 **+0.13** | 0.73 **+0.36** | 0.76 **+0.24** |
| | | | | *Autoregressive* | | | |
| Emu3 | 0.70 **+0.36** | 0.62 **+0.17** | 0.60 **+0.12** | 0.59 **+0.18** | 0.56 **+0.11** | 0.52 **+0.25** | 0.63 **+0.24** |
| Janus-Pro-7B | 0.75 **+0.45** | 0.66 **+0.29** | 0.70 **+0.21** | 0.71 **+0.35** | 0.73 **+0.31** | 0.59 **+0.33** | 0.71 **+0.36** |
| Janus-Pro-1B | 0.60 **+0.40** | 0.59 **+0.31** | 0.59 **+0.14** | 0.66 **+0.42** | 0.63 **+0.31** | 0.58 **+0.42** | 0.60 **+0.34** |
| Janus-1.3B | 0.40 **+0.24** | 0.48 **+0.22** | 0.49 **+0.14** | 0.54 **+0.26** | 0.53 **+0.23** | 0.44 **+0.30** | 0.46 **+0.23** |
| vila-u-7b-256 | 0.54 **+0.28** | 0.51 **+0.18** | 0.49 **+0.12** | 0.57 **+0.22** | 0.56 **+0.17** | 0.58 **+0.35** | 0.54 **+0.23** |

## 4.3. Evaluation on WISE Rewritten Prompts.

We further evaluate models on rewritten WISE prompts, where GPT-4o-2024-05-13 converts complex prompts into direct ones (e.g., "Common plants for Mother's Day" to "Carnation"). Details are in Appendix F, and results under the same WiScore protocol are shown in Table 2.

Nearly all models improve substantially after rewriting, with Qwen-Image reaching the highest overall WiScore of 0.88. This indicates that part of the original WISE difficulty comes from **models' inability to parse and contextualize complex, knowledge-intensive prompts**. By making the target entity or relation more explicit, prompt simplification removes this comprehension bottleneck and lets models better expose their inherent generative capability.

The largest gains tend to occur for models with lower original scores, suggesting that weaker models are more sensitive to indirect wording and contextualization difficulty. For example, Janus-Pro-1B improves by **+0.34**. BAGEL on rewritten prompts (0.73) is also close to BAGEL+CoT on the *original* prompts (0.70), suggesting that CoT partly acts as prompt simplification by eliciting and integrating internal knowledge during generation.

## 5. Conclusion

To evaluate whether current T2I and unified models can generate images beyond simple bag-of-words mappings, we introduced WISE, a benchmark of 1,000 questions across different domains designed to challenge models with common world knowledge. We evaluated 20 generative models, including dedicated T2I models and unified multimodal models, and found clear weaknesses in their ability to use world knowledge during image generation, especially among dedicated T2I models. Even for unified multimodal models, despite their strong language understanding capabilities, most open-source models cannot fully translate this advantage into strong image generation performance in complex and knowledge-intensive scenarios. Our analysis pinpoints the critical bottleneck: the challenge is not merely image quality, but the models' deep understanding and reasoning over complex prompts. The rewritten-prompt and CoT results further suggest that better prompt interpretation, knowledge elicitation, and tighter understanding-generation integration are important directions for future generative systems. Overall, WISE exposes the limitations of existing generative models from a benchmark perspective and highlights a promising direction for future methods.

## Acknowledgements

This work was supported in part by The Guangdong Grants (Grant No.2023ZT10X075), the Natural Science Foundation of China (No. 62332002, 62425101), and Shenzhen Science and Technology Program (KQTD20240729102051063).

## Impact Statement

This paper presents WISE, a benchmark aimed at advancing the field of Machine Learning by enhancing the world knowledge integration and complex semantic understanding of text-to-image models. By providing a structured evaluation framework across natural sciences and cultural domains, our work supports the development of more factually grounded and intelligent generative systems. While we acknowledge that generative models can reflect biases in their training data, our benchmark is designed to identify and bridge the "understanding-generation gap," ultimately promoting the creation of AI that is more reliable and better aligned with real-world facts.

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

*Table 3.* Prompt sourcing and quality-control summary for WISE. Initial candidates were gathered from diverse knowledge sources and then manually refined to ensure that each prompt is visually answerable, knowledge-intensive, and associated with explicit expected visual cues.

| Source / stage | Role in WISE construction | Quality-control action |
|---|---|---|
| Educational materials | Seed scientific concepts and school-level facts in biology, physics, and chemistry. | Kept concepts with clear visual manifestations; removed cases that mainly require equations, text reading, or non-visual answers. |
| Encyclopedias and public references | Provide factual entities, landmarks, cultural conventions, and geography-related associations. | Preferred stable and widely recognized knowledge; cross-checked ambiguous facts and avoided overly obscure or rapidly changing targets. |
| Common-sense problem sets | Supply everyday reasoning cases, spatial relations, temporal cues, and non-literal associations. | Converted answer-oriented facts into image-generation prompts; filtered prompts with multiple equally plausible visual targets. |
| LLM-assisted synthesis | Diversify candidate prompts and generate implicit phrasings across the 25 subdomains. | Human annotators rewrote or discarded candidates to maintain one inferable target, natural wording, and appropriate difficulty. |
| Human refinement | Finalize the benchmark prompts and attach explanations of the required world knowledge. | Checked clarity, category fit, visual answerability, and consistency between the prompt, expected answer, and explanation. |

*Table 4.* Cultural distribution audit. WISE is not dominated by one cultural bloc. The full benchmark is majority global/neutral, and the Cultural Common Sense subset is balanced between Western and non-Western prompts.

| Culture group | Full WISE | Cultural subset |
|---|---|---|
| Global / neutral | 56.6% | 17.75% |
| Western | 22.6% | 41.50% |
| Non-Western | 20.8% | 40.75% |

## A. WISE Category Descriptions

WISE encompasses a broad spectrum of knowledge categories, categorized under three main domains: Cultural Common Sense, Spatio-Temporal Reasoning, and Natural Science. Each domain is further divided into specific subcategories to assess different facets of understanding.

### A.1. Cultural Common Sense (400 prompts)

This domain evaluates the understanding of widely shared cultural knowledge and conventions. It includes:

To audit cultural coverage, we group prompts into global/neutral, Western, and non-Western categories. As shown in Table 4, the full benchmark is majority global/neutral, and the Cultural Common Sense subset is balanced between Western and non-Western prompts.

- **Festival:** This category assesses knowledge related to cultural celebrations, encompassing traditional foods, customs, and activities associated with specific festivals. *Example: Traditional cuisine of the Mid-Autumn Festival.*

- **Sports:** This category focuses on recognizing sports that are highly representative or culturally significant within particular nations or regions. *Example: The most representative sport of South Africa.*

- **Religion-related:** This category examines the identification of objects, symbols, or architectural structures that hold religious significance and are associated with specific faiths or religious heritage. *Example: A geometric symbol commonly associated with Jewish identity and heritage.*

- **Craft-related:** This category pertains to the recognition of traditional crafts that are emblematic of a nation's artistic and technical skills. *Example: A craft embodying Swiss precision and artistry.*

- **Construction-related:** This category tests the ability to identify iconic architectural structures that are representative landmarks of specific countries or cities. *Example: A sail-like structure, an architectural icon on Sydney's harbor.*

- **Animal:** This category assesses knowledge of animals that are symbolic, nationally significant or possess distinctive characteristics. *Example: A large animal, symbolizing national pride in Thailand.*

- **Plant:** This category evaluates the recognition of plants or fruits that are culturally representative, possess notable properties, or hold symbolic meaning. *Example: A famous flower symbolizing wealth in China.*

- **Art:** This category focuses on understanding artistic styles, recognizing representative musical instruments of different cultures, or identifying instruments with specific functional attributes. *Example: A triangular stringed instrument often used in Russian folk music.*

- **Celebrity:** This category assesses knowledge of globally recognized figures, including historical personalities, fictional characters from literature, or iconic roles from film and television. *Example: The iconic hat of the protagonist of One Piece in Japan.*

- **Life:** This category examines the understanding of common, everyday knowledge related to daily life practices and tools. *Example: A common utensil used for consuming steak in Western culture.*

## A.2. Spatio-Temporal Reasoning (300 prompts)

This domain evaluates the ability to reason about spatial and temporal relationships and contexts. It is further divided into Time and Space subcategories.

### A.2.1. TIME (167 PROMPTS)

This subsection focuses on temporal reasoning. It includes:

- **Horizontal Time:** This category assesses the understanding of temporal relationships across different entities or events occurring concurrently. *Example: The Pyramids of Giza at 8 PM Tokyo time.*

- **Longitudinal Time:** This category focuses on understanding temporal progression and order of events across time, including diurnal cycles, seasonal changes, and historical periods. *Example: The signature instrument of the rock and roll era in the 1950s.*

### A.2.2. SPACE (133 PROMPTS)

This subsection focuses on spatial reasoning. It includes:

- **Geographical Location:** This category examines the understanding of spatial relationships between geographical entities, such as cities, countries, and continents. *Example: A typical beverage produced in the country where Bordeaux is located.*

- **Relative Position:** This category assesses the ability to understand and interpret relative spatial positioning between objects, such as proximity, vertical placement, and size comparisons. *Example: A bird and a dog, with the smaller animal positioned on top of and the larger animal below.*

- **Different View:** This category evaluates the ability to recognize and interpret objects and scenes from various perspectives, including viewpoints like top-down, bottom-up, cross-sectional, side, mirrored, and occluded views. *Example: A view of a dense forest from within the canopy, looking upwards.*

## A.3. Natural Science (300 prompts)

This domain evaluates understanding of fundamental principles and phenomena within the natural sciences, categorized into Biology, Physics, and Chemistry.

A.3.1. BIOLOGY (100 PROMPTS)

- **State:** This subcategory assesses knowledge of the different physiological or developmental states of living organisms under varying conditions or across their life cycle stages. *Example: A maple tree with leaves exhibiting chlorophyll breakdown.*

- **Behavior:** This subcategory evaluates the understanding of typical behaviors exhibited by organisms, including actions related to survival, reproduction, and interaction with their environment. *Example: A morning rooster, emphasizing its characteristic behavior.*

A.3.2. PHYSICS (100 PROMPTS)

- **Mechanics:** This subcategory covers principles of mechanics, including:

  - **Gravity:** Understanding effects of gravity on objects, such as vertical suspension and equilibrium. *Example: A balloon filled with helium in a room.*
  - **Buoyancy:** Understanding the behavior of objects in fluids based on buoyancy principles.
  - **Pressure:** Understanding the effects of pressure and its variations.
  - **Surface Tension:** Understanding phenomena related to surface tension in liquids.
  - **Other Mechanical Phenomena:** Including concepts like wind effects on objects.

- **Thermodynamics:** This subcategory covers principles of heat and energy transfer, including:

  - **Evaporation:** Understanding the process of vaporization at boiling points.
  - **Liquefaction:** Understanding the condensation of gases into liquids.
  - **Solidification:** Understanding the process of freezing.
  - **Melting:** Understanding the process of fusion.
  - **Sublimation:** Understanding the phase transition from solid to gas.
  - **Deposition:** Understanding the phase transition from gas to solid. *Example: A pond at minus ten degrees Celsius.*

- **Optics:** This subcategory covers principles of light and vision, including:

  - **Refraction:** Understanding the bending of light as it passes through different media.
  - **Magnification:** Understanding how lenses magnify objects.
  - **Dispersion:** Understanding the separation of light into its spectral components. *Example: A laser beam passing through a dusty room.*

- **Physical Properties:** This subcategory assesses knowledge of material properties like electrical conductivity. *Example: An electrical bulb connected to a battery via copper wires.*

A.3.3. CHEMISTRY (100 PROMPTS)

- **Combustion:** This subcategory covers principles of chemical reactions involving rapid oxidation, including flame characteristics and color reactions. *Example: Lithium burning, highlighting its characteristic flame color.*

- **Metal Corrosion:** This subcategory assesses the understanding of the electrochemical degradation of metals over time due to environmental exposure. *Example: An iron block exhibiting rust due to corrosion.*

- **Solution Chemical Reaction:** This subcategory covers various types of chemical reactions in solutions, including acid-base reactions, redox reactions, and precipitation reactions. *Example: A T-shirt stained by sulfuric acid.*

- **Chemical Properties:** This subcategory assesses knowledge of intrinsic chemical properties, including colloidal behavior, protein chemistry, and the characteristic colors of chemical species in solution. *Example: A clear solution of copper sulfate exhibiting its characteristic color.*

*Table 5.* WiScore sensitivity to metric weights evaluated with gemini-3.1-flash-lite-preview. We compare the original consistency/realism/aesthetic weights (0.7/0.2/0.1) with two alternatives. Rankings are nearly unchanged; the Spearman correlation with the original setting is 0.993 for each alternative, and the correlation between the two alternatives is 1.000.

| Model | (0.7/0.2/0.1) | (0.5/0.3/0.2) | (0.4/0.4/0.2) |
|---|---|---|---|
| Qwen-Image | #1 (0.50) | #1 (0.52) | #1 (0.53) |
| UniWorld-V1 | #2 (0.44) | #2 (0.46) | #2 (0.46) |
| FLUX.1-dev | #3 (0.43) | #3 (0.45) | #2 (0.46) |
| SD-3.5-large | #4 (0.40) | #4 (0.41) | #4 (0.42) |
| SD-3.5-medium | #5 (0.37) | #5 (0.39) | #5 (0.40) |
| SD-3-medium | #6 (0.36) | #7 (0.37) | #7 (0.37) |
| SD-XL-base-0.9 | #7 (0.36) | #6 (0.38) | #6 (0.38) |
| FLUX.1-schnell | #8 (0.34) | #8 (0.35) | #8 (0.36) |
| Janus-Pro-7B | #9 (0.30) | #9 (0.31) | #9 (0.31) |
| SD-v1-5 | #10 (0.28) | #10 (0.28) | #10 (0.27) |
| SD-2-1 | #11 (0.27) | #11 (0.28) | #11 (0.27) |
| Janus-Pro-1B | #12 (0.23) | #12 (0.23) | #12 (0.22) |
| Janus-1.3B | #13 (0.20) | #13 (0.19) | #13 (0.18) |

*Table 6.* WiScore ranking stability under different VLM judges. The representative model ranking remains stable when replacing the original GPT-4o judge with gemini-3.1-flash-lite-preview or Qwen3.5-122B-A10B. Each cell reports rank and score.

| Model | Original judge | gemini-3.1-flash-lite-preview | Qwen3.5-122B-A10B |
|---|---|---|---|
| Qwen-Image | #1 (0.62) | #1 (0.50) | #1 (0.57) |
| UniWorld-V1 | #2 (0.55) | #2 (0.44) | #2 (0.45) |
| FLUX.1-dev | #3 (0.50) | #3 (0.43) | #2 (0.45) |
| SD-3.5-large | #4 (0.46) | #4 (0.40) | #4 (0.44) |
| SD-3.5-medium | #5 (0.45) | #5 (0.37) | #5 (0.42) |
| SD-XL-base-0.9 | #6 (0.43) | #6 (0.36) | #6 (0.40) |
| SD-3-medium | #7 (0.42) | #6 (0.36) | #6 (0.40) |
| FLUX.1-schnell | #8 (0.40) | #8 (0.34) | #9 (0.37) |
| Janus-Pro-7B | #9 (0.35) | #9 (0.30) | #8 (0.38) |
| SD-v1-5 | #10 (0.32) | #10 (0.28) | #10 (0.34) |
| SD-2-1 | #10 (0.32) | #11 (0.27) | #10 (0.34) |
| Janus-Pro-1B | #12 (0.26) | #12 (0.23) | #13 (0.28) |
| Janus-1.3B | #13 (0.23) | #13 (0.20) | #12 (0.29) |

## B. WiScore Weight Sensitivity

WiScore assigns the largest weight to consistency because WISE primarily evaluates whether a generated image satisfies the knowledge constraints implied by the prompt. To verify that this design choice does not determine the model ranking, we evaluate two alternative weighting schemes with higher realism and aesthetic weights using gemini-3.1-flash-lite-preview as the VLM evaluator. As shown in Table 5, rankings remain nearly unchanged: the Spearman rank correlation between the original weights and each alternative is 0.993, and the correlation between the two alternatives is 1.000.

## C. VLM Judge Stability

To reduce dependence on a single closed-source evaluator, we also evaluate representative models with gemini-3.1-flash-lite-preview and Qwen3.5-122B-A10B. As shown in Table 6, model rankings are stable across evaluators, with only minor adjacent swaps among close models.

## D. Scoring Criteria

The detailed scoring criteria are shown in Figure 5.

---

**Scoring Criteria**

**Consistency (0-2): How accurately and completely the image reflects the PROMPT.**
**0 (Rejected):** Fails to capture key elements of the prompt, or contradicts the prompt.
**1 (Conditional):** Partially captures the prompt. Some elements are present, but not all, or not accurately. Noticeable deviations from the prompt's intent.
**2 (Exemplary):** Perfectly and completely aligns with the PROMPT. Every single element and nuance of the prompt is flawlessly represented in the image. The image is an ideal, unambiguous visual realization of the given prompt.

**Realism (0-2): How realistically the image is rendered.**
**0 (Rejected):** Physically implausible and clearly artificial. Breaks fundamental laws of physics or visual realism.
**1 (Conditional):** Contains minor inconsistencies or unrealistic elements. While somewhat believable, noticeable flaws detract from realism.
**2 (Exemplary):** Achieves photorealistic quality, indistinguishable from a real photograph. Flawless adherence to physical laws, accurate material representation, and coherent spatial relationships. No visual cues betraying AI generation.

**Aesthetic Quality (0-2): The overall artistic appeal and visual quality of the image.**
**0 (Rejected):** Poor aesthetic composition, visually unappealing, and lacks artistic merit.
**1 (Conditional):** Demonstrates basic visual appeal, acceptable composition, and color harmony, but lacks distinction or artistic flair.
**2 (Exemplary):** Possesses exceptional aesthetic quality, comparable to a masterpiece. Strikingly beautiful, with perfect composition, a harmonious color palette, and a captivating artistic style. Demonstrates a high degree of artistic vision and execution.

*Figure 5.* WISE assesses image quality using three criteria: how accurately the image aligns with the prompt (Consistency), its level of realism (Realism), and its overall artistic appeal (Aesthetic Quality). Each metric is scored on a scale from 0 (Rejected) to 2 (Exemplary), providing an evaluation of image fidelity, believability, and visual quality.

## E. GPT-4o Assessment Instruction

Figure 6 presents the instructions provided to GPT-4o for evaluation.

## F. Using GPT-4o to Rewrite WISE Prompts

Figure 7 presents the instructions provided to GPT-4o for prompt rewriting.

### F.1. Rewrite Gain Analysis

We further analyze whether the gains from prompt rewriting primarily reflect rewrite-removable prompt comprehension difficulty or persistent knowledge-informed generation difficulty. For efficiency and cost reasons, this analysis uses gemini-3.1-flash-lite-preview as the VLM evaluator. We compute the across-model mean rewriting gain for each prompt and also use a threshold-based partition over originally difficult prompts. As shown in Table 7, rewriting helps many cases, but 60.1% of originally difficult prompts remain rewrite-ineffective. The pattern is category-dependent: Cultural Common Sense has a larger rewrite-effective portion, while scientific categories, especially Physics, remain difficult after rewriting. Across multiple thresholds, the rewrite-ineffective subset consistently remains larger and its ranking remains highly correlated with the overall ranking.

## G. Failure Case Analysis

We further inspect representative failure cases to understand what types of knowledge are difficult for current T2I models. As summarized in Table 8, errors mainly fall into three categories: missing implicit associations, violations of scientific constraints, and inaccurate visualization of fine-grained states. These failures are not fully resolved by prompt rewriting. For

# Text-to-Image Quality Evaluation Protocol
## System Instruction
You are an AI quality auditor for text-to-image generation. Apply these rules with ABSOLUTE RUTHLESSNESS. Only images meeting the HIGHEST standards should receive top scores.
**Input Parameters**
- PROMPT: [User's original prompt to]
- EXPLANATION: [Further explanation of the original prompt]
---
## Scoring Criteria
**Consistency (0-2):** How accurately and completely the image reflects the PROMPT.
* **0 (Rejected):** Fails to capture key elements of the prompt, or contradicts the prompt.
* **1 (Conditional):** Partially captures the prompt. Some elements are present, but not all, or not accurately. Noticeable deviations from the prompt's intent.
* **2 (Exemplary):** Perfectly and completely aligns with the PROMPT. Every single element and nuance of the prompt is flawlessly represented in the image. The image is an ideal, unambiguous visual realization of the given prompt.
**Realism (0-2):** How realistically the image is rendered.
* **0 (Rejected):** Physically implausible and clearly artificial. Breaks fundamental laws of physics or visual realism.
* **1 (Conditional):** Contains minor inconsistencies or unrealistic elements. While somewhat believable, noticeable flaws detract from realism.
* **2 (Exemplary):** Achieves photorealistic quality, indistinguishable from a real photograph. Flawless adherence to physical laws, accurate material representation, and coherent spatial relationships. No visual cues betraying AI generation.
**Aesthetic Quality (0-2):** The overall artistic appeal and visual quality of the image.
* **0 (Rejected):** Poor aesthetic composition, visually unappealing, and lacks artistic merit.
* **1 (Conditional):** Demonstrates basic visual appeal, acceptable composition, and color harmony, but lacks distinction or artistic flair.
* **2 (Exemplary):** Possesses exceptional aesthetic quality, comparable to a masterpiece. Strikingly beautiful, with perfect composition, a harmonious color palette, and a captivating artistic style. Demonstrates a high degree of artistic vision and execution.
---
## Output Format
**Do not include any other text, explanations, or labels.** You must return only three lines of text, each containing a metric and the corresponding score, for example:
**Example Output:**
Consistency: 2
Realism: 1
Aesthetic Quality: 0
---
**IMPORTANT Enforcement:**
Be EXTREMELY strict in your evaluation. A score of '2' should be exceedingly rare and reserved only for images that truly excel and meet the highest possible standards in each metric. If there is any doubt, downgrade the score.
For **Consistency**, a score of '2' requires complete and flawless adherence to every aspect of the prompt, leaving no room for misinterpretation or omission.
For **Realism**, a score of '2' means the image is virtually indistinguishable from a real photograph in terms of detail, lighting, physics, and material properties.
For **Aesthetic Quality**, a score of '2' demands exceptional artistic merit, not just pleasant visuals.

*Figure 6.* We use GPT-4o to evaluate the performance of text-to-image models. The figure shows the instruction provided to GPT-4o.

example, Chemistry still has a 74.0% rewrite-ineffective ratio in Table 7, suggesting that many errors arise from mapping chemical states and material properties into images, rather than only from understanding indirect wording.

*Table 7.* Prompt rewriting gain analysis evaluated with gemini-3.1-flash-lite-preview. Prompts are partitioned by the across-model change from the original WISE prompt to the rewritten prompt. "Effective" means rewriting resolves an originally difficult prompt, while "Ineffective" means the prompt remains difficult after rewriting.

| Category | Rewrite-effective ratio | Rewrite-ineffective ratio |
|---|---|---|
| Cultural Common Sense | 56.8% | 43.2% |
| Time | 37.6% | 62.4% |
| Space | 30.8% | 69.2% |
| Biology | 28.9% | 71.1% |
| Physics | 14.9% | 85.1% |
| Chemistry | 26.0% | 74.0% |
| Originally difficult prompts | 39.9% | 60.1% |

| Threshold | Ratio (Eff. / Ineff.) | Spearman (Eff. / Ineff.) |
|---|---|---|
| 0.6 | 34.3% / 51.6% | 0.9273 / 1.0000 |
| 0.7 | 24.8% / 66.7% | 0.9636 / 0.9909 |
| 0.8 | 13.6% / 83.2% | 0.9182 / 0.9909 |

*Table 8.* Representative failure patterns on WISE. These cases show that errors arise not only from surface prompt parsing, but also from missing associative knowledge, violated scientific constraints, and inaccurate visual mapping of fine-grained states.

| Failure pattern | Typical model behavior | Representative WISE case |
|---|---|---|
| Implicit association missing | The model recognizes words in the prompt but fails to retrieve the real-world association needed to identify the visual target. | For prompts such as "common plants for Mother's Day," models may generate generic flowers instead of the culturally associated target, carnation. |
| Scientific constraint violation | The generated image is visually plausible but contradicts physical or biological constraints implied by the scenario. | For a candle in outer space, models may still render a normal burning flame, ignoring the oxygen-dependent condition for combustion. |
| Fine-grained state confusion | The model retrieves the broad entity but fails to map a specific material state, reaction stage, or structural relation into correct visual details. | For galvanized steel with early moisture-induced corrosion, models often produce generic red rust rather than localized white corrosion products linked to zinc coating damage. |

*Table 9.* Multi-seed stability on WISE evaluated with gemini-3.1-flash-lite-preview. We generate images with five random seeds (42–46) and report mean, standard deviation, 95% confidence interval, and the rank range across seeds. The overall ranking is stable, with only minor adjacent swaps among close models.

| Model | Mean $\pm$ Std | 95% CI | Rank range |
|---|---|---|---|
| Qwen-Image | $0.5029 \pm 0.0046$ | [0.4972, 0.5086] | 1–1 |
| FLUX.1-dev | $0.4225 \pm 0.0045$ | [0.4168, 0.4281] | 2–2 |
| SD-3.5-large | $0.4040 \pm 0.0092$ | [0.3926, 0.4154] | 3–3 |
| SD-3.5-medium | $0.3714 \pm 0.0015$ | [0.3696, 0.3733] | 4–4 |
| SD-3-medium | $0.3584 \pm 0.0042$ | [0.3532, 0.3636] | 5–6 |
| SD-XL-base-0.9 | $0.3584 \pm 0.0087$ | [0.3476, 0.3692] | 5–7 |
| FLUX.1-schnell | $0.3388 \pm 0.0231$ | [0.3102, 0.3675] | 5–7 |
| Janus-Pro-7B | $0.3043 \pm 0.0014$ | [0.3026, 0.3060] | 8–8 |
| SD-v1-5 | $0.2758 \pm 0.0084$ | [0.2654, 0.2862] | 9–9 |
| Janus-Pro-1B | $0.2339 \pm 0.0028$ | [0.2305, 0.2374] | 10–10 |
| Janus-1.3B | $0.2080 \pm 0.0054$ | [0.2013, 0.2147] | 11–11 |

# H. Multi-seed Stability

To account for stochasticity in image generation, we run representative models with five random seeds (42–46) and evaluate them with gemini-3.1-flash-lite-preview for efficiency. As shown in Table 9, the confidence intervals are narrow and the rank ranges are stable, indicating that seed variance does not affect the main comparative conclusions.

*Table 10.* Human evaluation protocol and inter-annotator agreement. Each image is scored by five independent annotators on consistency, realism, and aesthetic quality. The final human score follows the same weighting scheme as WiScore. Annotators were undergraduate- or PhD-level participants and were allowed to use search engines for factual verification.

| Item | Setting |
|------|---------|
| Models evaluated | Qwen-Image and UniWorld |
| Images per model | 180 images |
| Category sampling | 80 Cultural Common Sense; 20 from each remaining category |
| Annotators per image | 5 independent annotators |
| Annotator background | Undergraduate- or PhD-level participants |
| Scoring dimensions | Consistency, realism, aesthetic quality |
| External verification | Search engines allowed for factual verification |
| Aggregation | Average over annotators; same weighted composition as WiScore |

| Dimension | Krippendorff's $\alpha$ |
|-----------|------------------------|
| Consistency | 0.82 |
| Realism | 0.78 |
| Aesthetic quality | 0.67 |

*Table 11.* Additional evaluation of recent AR and AR+Diffusion models. Recent AR models substantially improve over early AR baselines, while AR+Diffusion models remain the strongest overall in this supplementary evaluation.

| Paradigm | Model | WiScore |
|----------|-------|---------|
| Autoregressive | LongCat-Next (Team et al., 2026) | 0.57 |
| Autoregressive | Infinity (Han et al., 2025) | 0.47 |
| Autoregressive | Emu3.5 (Cui et al., 2025) | 0.57 |
| Autoregressive | NextFlow (Zhang et al., 2026) | 0.62 |
| AR+Diffusion | LongCat-Image (Team et al., 2025) | 0.65 |
| AR+Diffusion | Hunyuan-Image 3.0 (Cao et al., 2025) | 0.57 |
| AR+Diffusion | DeepGen1.0 (Wang et al., 2026) | 0.73 |

## I. Human Evaluation Details

We summarize the human annotation protocol used to validate WiScore. Each image is evaluated by five independent annotators along the same three dimensions used by WiScore: consistency, realism, and aesthetic quality. The final human score is computed with the same weighting scheme as WiScore by averaging over annotators. Annotators were undergraduate- or PhD-level participants. We did not require specific disciplinary expertise, but annotators were allowed to use search engines to verify factual requirements for knowledge-intensive cases.

## J. Additional Architecture Evaluation

To further assess whether the architecture-level observations depend on relatively early unified models, we additionally evaluate recent autoregressive and AR+Diffusion models in Table 11. Recent AR models show clear progress over the early AR baselines in Table 1: LongCat-Next (Team et al., 2026), Emu3.5 (Cui et al., 2025), and NextFlow (Zhang et al., 2026) reach substantially higher WiScore than Janus, VILA-U, and Emu3. Nevertheless, the AR+Diffusion models remain stronger overall in this supplementary evaluation, with LongCat-Image (Team et al., 2025) and DeepGen1.0 (Wang et al., 2026) outperforming the recent AR group. These results suggest that stronger AR backbones reduce the gap, but tighter coupling between language-level reasoning and diffusion-based visual generation remains beneficial for WISE.

## K. Resolution Confound Check

To examine whether model resolution confounds the WISE ranking, we re-evaluated representative models from the Qwen-Image, FLUX, and Stable Diffusion families under a unified $512 \times 512$ resolution setting. The overall ranking trend remains unchanged; the only observed change is a minor adjacent swap where SD-3.5-medium exceeds SD-3.5-large by 0.01. This suggests that image resolution is not the primary driver of the comparative conclusions on WISE.

You are a prompt rewriting assistant. Your task is to transform complex prompts into direct, image-focused prompts suitable for a text-to-image model. You will receive a Prompt and an Explanation. Rewrite the Prompt to clearly describe the image to be generated, incorporating relevant details from the Explanation. The rewritten prompt should be self-contained and not require the Explanation to understand.

*Examples:*

**Prompt:** A famous flower that symbolizes wealth in China.

**Explanation:** This refers to the peony, often called the 'King of Flowers' in China, symbolizing wealth, prosperity, and good fortune in Chinese culture.

**Output:** Peony.

---

**Prompt:** A solution of silver nitrate before light exposure.

**Explanation:** The model should generate an image depicting a clear, colorless liquid in a transparent container. There should be no visible precipitate or cloudiness, representing a stable silver nitrate solution protected from light.

**Output:** An image depicting a clear, colorless liquid in a transparent container. There should be no visible precipitate or cloudiness, representing a stable silver nitrate solution protected from light.

---

**Prompt:** The currency of the largest country by area in the world.

**Explanation:** The model should generate an image of the Russian Ruble.

**Output:** Russian Ruble.

---

*Figure 7.* We use GPT-4o to rewrite prompts in WISE, transforming complex, knowledge-demanding prompts into direct prompts. The figure shows the instruction provided to GPT-4o.

## L. Limitations

While WISE evaluates the complex semantic understanding **(implicit understanding)** and world knowledge **(intrinsic knowledge matching)** capabilities of dedicated T2I and unified models, we acknowledge several limitations. Our benchmark categorizes prompts into broad domains, but due to the interconnected nature of knowledge, some prompts may inherently span multiple categories (e.g., "the impact of climate change on polar bear habitats" could fall under both natural science and spatio-temporal reasoning), potentially introducing ambiguity in cross-category analysis. Furthermore, while WISE covers a range of topics, it represents a sample of knowledge domains and cannot encompass all aspects of world knowledge, which is also constantly evolving. Additionally, some models were not publicly available or did not provide APIs at the time of our work's deadline, precluding their evaluation in this study. Finally, WISE currently focuses on static single-image generation. Extending it to multi-image or interactive generation would require sequence-level annotations and cross-image consistency metrics, which we leave for future work.

