# OpenReview forum: "WISE: World Knowledge-Informed Semantic Evaluation for Text-to-Image Generation"
_ICML.cc/2026/Conference — ICML 2026 regular_

### Official Review · Reviewer_QvBt · 2026-02-14

**Soundness:** 3
**Presentation:** 3
**Significance:** 3
**Originality:** 3
**Overall Recommendation:** 5
**Confidence:** 5

**Summary:**

This paper addresses a key and previously under-looked dimension in text-to-image generation analysis and investigates the content to which contemporary generation systems perceive and manipulate "world knowledge". To this end, the authors curated a novel benchmark (WISE) and proposed a new evaluation metric (WiScore) and conducted a systematic evaluation for different contemporary generation systems (diffusion-based and different unified multimodal models). Additional to quantitative evaluation scores, the authors gave further insights regarding the effect of different architecture designs, different types of parsing prompts and other limitations of current models.

**Compliance With Llm Reviewing Policy:**

Affirmed.

**Final Justification:**

This paper addresses a key and previously under-looked dimension in text-to-image generation analysis and investigates the content to which contemporary generation systems perceive and manipulate "world knowledge". The authors appropriately addressed the raised concerns and added more systematic evaluations in the rebuttal, therefore I believe that this paper should be accepted.

**Key Questions For Authors:**

See weaknesses.

**Limitations:**

Yes.

**Strengths And Weaknesses:**

Strengths:
1. This paper is well grounded and addresses an important factor in evaluating generation systems. The proposed benchmark serves as a solid contribution for evaluating the inherent "world knowledge" understanding for generation systems.
2. Overall presentation of this paper is clear and easy to follow.

Weaknesses:
Overall, some of the claims in this paper are questionable.
1. Certain claims should be supported by more evaluations: The authors investigated the impact of different model architectures in Section 4.2. This should be supported by a more comprehensive evaluation on more state-of-the-art models, especially for autoregressive generation models, which the authors only presented results for relatively old models. The authors should consider including stronger, more recent models [1,2]. This is important because the inclusion of more up-to-date models could change/impact the observations that are made regarding different architectures.
2. Definition of generation paradigms: The definition of "Cascaded AR-Diffusion models" adopted by the authors seems a little strange. Autoregressive + diffusion models is usually termed as "language backbone autoregressively generates a set of latent tokens as 'semantic condition' signals, which are then employed by the diffusion module to generate images" [3]. Qwen-Image should not be viewed as AR-Diffusion models under this approach.
3. Lack of failure case analysis: The authors could benefit from conducting a more thorough analysis on the failure cases produced by models on the proposed benchmarks. Analyzing the failure patterns (what type of knowledge is absent?), the hard cases (before & after prompt rewriting) would make the paper more solid and insightful.
4. Unclear metric formulation: it seems unclear how the weights for different dimensions are formed, given that the values are untrivial.

[1] Cui, Yufeng, et al. "Emu3. 5: Native multimodal models are world learners."

[2] Han, Jian, et al. "Infinity: Scaling bitwise autoregressive modeling for high-resolution image synthesis."

[3] Deng, Chaorui, et al. "Emerging properties in unified multimodal pretraining."

---

> ### Author Rebuttal · Authors · 2026-03-31
>
> **W1**: Architectural Comparison Needs More Recent Model Support
>
> **A:** These results indicate that recent stronger autoregressive models have also achieved impressive performance on WISE, suggesting that the AR paradigm has strong potential to leverage world knowledge for visual generation. We will cite and discuss these excellent works in the revised version.
>
> **W2**: The Generation Paradigm Definition of Qwen-Image
>
> **A:** Thank you for your suggestion. We would like to clarify that our use of "Cascaded AR-Diffusion" is based on an implementation-oriented definition: we explicitly describe it in the paper as "using the AR model's last hidden state to pass condition," meaning that the upstream AR/MLLM module first performs semantic modeling and then passes its hidden states as conditioning signals to the diffusion generator. Qwen-Image does exactly this by passing the last hidden state of its AR model (Qwen) as a condition to the Diffusion Model, which is why we group Qwen-Image, UniWorld, BLIP3o, and MetaQuery together.
>
> At the same time, we understand the reviewer's concern: in stricter usage, "AR-Diffusion" often specifically refers to AR backbones that explicitly autoregressively generate latent/semantic tokens, which are then used by the diffusion module to produce images. In the revised version, we will rename this category to a more accurate term, such as **"AR-conditioned diffusion,"** to more clearly emphasize that we are distinguishing the connection mechanism between the AR/MLLM and the diffusion model, rather than referring strictly to explicit AR token generation.
>
> **W3**: Failure Case Analysis
>
> **A:** Thank you for your suggestion. We will add a dedicated Failure Case Analysis section in the revised version. In practice, knowledge deficiencies in failure cases come from multiple types. For example, models may recognize individual words but lack the implicit associative knowledge that connects entities in the real world (e.g., the relationship between Mother's Day and carnations). Another common type is the absence of scientific constraint knowledge, where models are asked to generate images in specific scenarios but ignore scientific rules—for instance, incorrectly generating a burning flame for "a candle in outer space."
>
> Furthermore, Chemistry is the lowest-scoring category both before and after prompt rewriting (0.297 before, 0.591 after), indicating that chemistry-related knowledge poses the greatest challenge for current T2I models. Although rewriting substantially alleviates the difficulty of parsing implicit knowledge cues, yielding an improvement of approximately +0.294 for Chemistry, its post-rewriting score remains the lowest among all categories. This suggests that the difficulty of chemistry-related tasks does not stem solely from complex prompt comprehension but also from deeper challenges in visualizing scientific knowledge.
>
> For example, even after rewriting "A piece of galvanized steel exposed to moisture, with early signs of corrosion" into "A piece of galvanized steel with spots of white corrosion products forming where the zinc coating is compromised, displaying early stages of oxidation," models may still fail to correctly render the key visual features of early-stage galvanized corrosion—instead generating generic rusted metal, incorrectly rendering large areas of red rust, or ignoring the correspondence between localized white corrosion products and coating damage. This indicates that the core bottleneck for chemistry-related examples lies not only in retrieving the target concept but also in the model's inability to accurately map fine-grained chemical states, reaction stages, and material properties into visual outputs.
>
> **W4**: WiScore Metric Weights
>
> **A:** Thank you for your suggestion. We would like to clarify that the current weights are primarily guided by the design principle of WISE, which emphasizes knowledge consistency, and therefore assigns the largest weight to the consistency dimension. We have conducted an additional weight sensitivity analysis by changing the weights from the original (0.7/0.2/0.1) to (0.5/0.3/0.2) and (0.4/0.4/0.2). The results show that the model rankings are nearly identical across the three settings: the Spearman rank correlation between the original weights and each alternative is 0.993, and the correlation between the two alternatives is 1.000. Except for one adjacent swap between stable-diffusion-3-medium and stable-diffusion-xl-base-0.9, all other model rankings remain unchanged. This demonstrates that our main conclusions are not sensitive to the weight choice.

---

> > ### Author Rebuttal · Reviewer_QvBt · 2026-04-01
> >
> > Thanks to the authors for the thorough rebuttal and my concerns have been mostly addressed. I believe this work serves as a technically solid contribution to the generation/unified modeling community. If the authors could provide further in depth analysis on architectural differences with updated evaluation results from SOTA AR models (see weakness 1) in the discussion, I would be willing to further raise my score.

---

> > > ### Author Response · Authors · 2026-04-01
> > >
> > > Thank you for the valuable suggestion, and we apologize for the confusion.
> > >
> > > In addition to **Infinity** and **Emu3.5**, we further extended our evaluation to include several **recent state-of-the-art autoregressive (AR) models** (LongCat-Next, NextFlow) as well as **latest AR+Diffusion models** (LongCat-Image, Hunyuan-Image 3.0, DeepGen1.0). The results are summarized below:
> > >
> > > ### Evaluation Results
> > >
> > > **Autoregressive (AR) models**
> > >
> > > | Model | WiScore |
> > > |------|--------|
> > > | LongCat-Next | 0.57 |
> > > | Infinity | 0.47 |
> > > | Emu3.5 | 0.57 |
> > > | NextFlow | 0.62 |
> > >
> > > **AR + Diffusion models**
> > >
> > > | Model | WiScore |
> > > |------|--------|
> > > | LongCat-Image | 0.65 |
> > > | Hunyuan-Image 3.0 | 0.57 |
> > > | DeepGen1.0 | 0.73 |
> > >
> > > It can be observed that AR+Diffusion models still represent the most effective architecture overall. However, recent advances in AR models enable them to match or even surpass certain AR+Diffusion models, demonstrating the strong potential of AR models in leveraging world knowledge for generation.
> > >
> > > We further analyze the architectural differences. The strong performance of AR+Diffusion models can be attributed to the fact that diffusion remains the dominant paradigm for high-quality image synthesis. By combining AR and diffusion, these models effectively integrate the strengths of both paradigms: AR provides strong planning and reasoning capabilities, while diffusion excels at high-fidelity image rendering.
> > >
> > > In contrast, the AR paradigm is more closely aligned with large language models (LLMs), following a unified autogressive prediction framework. This alignment endows AR models with strong reasoning potential and highly efficient parallel training. As a result, recent progress in AR models demonstrates that they are a promising direction for knowledge-informed generation.
> > >
> > > We sincerely thank the reviewer for the valuable suggestion. We will include these findings and cite these excellent works in the revised manuscript. Should you have any further recommendations for additional AR models, we would be more than happy to incorporate them.

---

### Official Review · Reviewer_3aNe · 2026-03-10

**Soundness:** 2
**Presentation:** 3
**Significance:** 2
**Originality:** 2
**Overall Recommendation:** 3
**Confidence:** 5

**Summary:**

This paper introduces WISE, a benchmark for evaluating the ability of text-to-image (T2I) models to incorporate world knowledge facts and reasoning when generating images. The benchmark contains 1,000 prompts across 25 subdomains, covering cultural common sense, spatio-temporal reasoning, and natural science.

**Compliance With Llm Reviewing Policy:**

Affirmed.

**Final Justification:**

I keep my Score.

**Key Questions For Authors:**

**1. Human evaluation protocol.**

Could the authors provide more details about the human evaluation process? Specifically:
- How were annotators instructed to evaluate images?
- Were the same criteria (consistency, realism, aesthetics) used?
- How many annotators evaluated each sample?
- Inter-annotator agreement.


**2. LLM judge stability.**

How sensitive are the model results to the choice of the LLM evaluator? For example, would the results change significantly if a different multimodal model were used as the judge or if different prompts were used for evaluation?

**Limitations:**

Yes, a limitation section is included

**Strengths And Weaknesses:**

### **Strengths**

The paper investigates an interesting question: whether current T2I models truly integrate world knowledge and reasoning skills rather than performing shallow text–image alignment. The proposed prompts cover multiple domains (culture, spatial reasoning, science), which go beyond many existing benchmarks focusing mainly on compositional attributes. The evaluation is comprehensive and covers most state-of-the-art models.

### **Weaknesses**


**1.  The evaluation on world knowledge/subjec knowledge reasoning is not entirely new.** Prior benchmarks have already explored similar aspects of compositionality and knowledge in text-to-image generation, such as:

- [2025 ICLR] ScImage: How Good Are Multimodal Large Language Models at Scientific Text-to-Image Generation?
- [2023 Neurips] T2I-CompBench: A Comprehensive Benchmark for Open-world Compositional Text-to-image Generation. The paper should clarify how WISE differs from these prior benchmarks.


**2. Limitation of the proposed metric (WiScore).**

The contribution of WiScore appears relatively limited compared with prior metrics such as CLIPScore or VQA-based evaluation as the author mentioned and compared. In practice, WiScore mainly combines three scores (consistency, realism, aesthetic quality) that are themselves obtained from a GPT-based evaluator, using a weighted average.

This makes the metric closer to a composition of LLM judgments rather than a fundamentally new evaluation methodology.

**3. Missing ground truth image reference.**

As a benchmark, the paper seems not provide any reference image as ground truth and for comparison with other model provided images.

**4. Insufficient description of human evaluation.**

The paper claims that WiScore aligns well with human judgments on 100 examples, but the details of the human evaluation protocol are unclear. It is not specified:
- how human annotators were instructed,
- what criteria and instructions are given when scoring images,
- whether annotators evaluated the same dimensions (consistency, realism, aesthetics).

Human evaluations may vary depending on what aspects annotators focus on (e.g., semantic correctness vs. visual quality). The paper does not report inter-annotator agreement or analyze potential variability in human judgments.

**5. Missing analysis of specific dimensions.**

Although WiScore aggregates multiple dimensions into a single number, it would also be useful to evaluate models separately on consistency, realism, and aesthetics using human annotations. Such analysis could provide clearer insight into where current models fail (e.g., knowledge reasoning vs. visual quality).

---

> ### Author Rebuttal · Authors · 2026-03-31
>
> **Q1 & W4**: Human Evaluation Protocol
>
> **A:** Thank you for this important suggestion. We provide the following details: each image is evaluated by 5 independent annotators; all annotators score each image along the three dimensions of consistency, realism, and aesthetics; detailed annotation instructions will be included in the appendix of the revised version due to space constraints; the final score is computed using the same weighting scheme as WiScore; and each sample's result is obtained by averaging all annotators' scores. Regarding annotator background, we will clarify in the revision: our annotators are at the undergraduate level or phd level. Although we did not screen annotators by specific disciplinary expertise, they were allowed to use search engines for verification.
> We have also expanded the human evaluation to 180 images each from Qwen-Image and UniWorld (80 from the cultural category and 20 from each of the remaining five categories), reducing the bias caused by single-model, small-sample validation. The following results are reported on the 180-image evaluation subset:
> | Model | WiScore | Human Score |
> |---|---|---|
> | Qwen-Image | 0.50 | 0.54 |
> |  UniWorld  | 0.40 | 0.42 |
>
> | Dimension | Krippendorff's α |
> |---|---|
> | Consistency | 0.82 |
> | Realism | 0.78 |
> | Aesthetics | 0.67 |
>
> The high inter-annotator agreement and close alignment between human and WiScore values demonstrate the reliability of our evaluation.
>
> **Q2**: LLM Judge Stability
>
> **A:** Thank you for your suggestion. We evaluated WiScore with Gemini 3.1 Flash and the open-source Qwen3.5-122B-A10B as alternative judges. Due to rebuttal-time and compute resources limits, we used representative models from the Qwen-Image, UniWorld, FLUX, Stable Diffusion, and Janus families. The overall ranking is stable across judges:
>
> | Model | Original | Gemini | Qwen |
> |---|---|---|---|
> | Qwen-Image | #1 (0.62) | #1 (0.50) | #1 (0.57) |
> | UniWorld-V1 | #2 (0.55) | #2 (0.44) | #2 (0.45) |
> | FLUX.1-dev | #3 (0.50) | #3 (0.43) | #2 (0.45) |
> | SD-3.5-large | #4 (0.46) | #4 (0.40) | #4 (0.44) |
> | SD-3.5-medium | #5 (0.45) | #5 (0.37) | #5 (0.42) |
> | SD-XL-base-0.9 | #6 (0.43) | #6 (0.36) | #6 (0.40) |
> | SD-3-medium | #7 (0.42) | #6 (0.36) | #6 (0.40) |
> | FLUX.1-schnell | #8 (0.40) | #8 (0.34) | #9 (0.37) |
> | Janus-Pro-7B | #9 (0.35) | #9 (0.30) | #8 (0.38) |
> | SD-v1-5 | #10 (0.32) | #10 (0.28) | #10 (0.34) |
> | SD-2-1 | #10 (0.32) | #11 (0.27) | #10 (0.34) |
> | Janus-Pro-1B | #12 (0.26) | #12 (0.23) | #13 (0.28) |
> | Janus-1.3B | #13 (0.23) | #13 (0.20) | #12 (0.29) |
>
> **W1**: Insufficient Differentiation from Existing Benchmarks
>
> **A:** We agree that prior work has begun to address compositional or domain-specific evaluation in T2I generation, but WISE targets a distinct aspect. T2I-CompBench evaluates explicit compositional constraints (attribute binding, object relations), and ScImage focuses on scientific diagram generation with explicit spatial/numeric constraints. In contrast, WISE addresses world-knowledge-informed semantic generation: prompts do not directly specify the target visual entity but require the model to perform implicit semantic parsing and world knowledge retrieval. Therefore, WISE evaluates knowledge-informed generation in open-world scenarios, rather than pure compositionality or scientific diagram generation.
>
> **W2**: On the Contribution of WiScore
>
> **A:** We agree that WiScore builds upon existing LLM-based evaluation paradigms. The core contribution of this paper is the WISE benchmark itself; WiScore is a task-specific companion metric. Existing metrics (CLIPScore, VQAScore) focus on shallow semantic matching and cannot assess world-knowledge consistency, whereas WiScore explicitly evaluates this. Our experiments show WiScore achieves the highest agreement with human evaluation among all tested metrics, validating its effectiveness.
>
> **W3**: Lack of Ground-truth / Reference Image
>
> **A:** WISE does not adopt reference-image-based evaluation because a single prompt can correspond to multiple valid but visually different images; forcing a single reference would shift the evaluation from "knowledge consistency" to "visual similarity." Instead, we provide a detailed Explanation for each prompt specifying the factual requirements the generated image should satisfy, serving as the "semantic ground truth." We believe this knowledge-constraint-based evaluation is more suitable for measuring complex semantic accuracy than a single reference image.
>
> **W5**: Lack of Dimension-wise Analysis Using Human Annotations
>
> **A:** We thank the reviewer for this suggestion. The per-dimension results show that consistency is the weakest dimension for both models (Uniworld: 0.77, Qwen: 0.96), while aesthetic quality is comparatively stronger (1.04 and 1.30, respectively). This suggests that the current bottleneck is not primarily visual quality, but knowledge consistency.

---

> > ### Author Rebuttal · Reviewer_3aNe · 2026-04-01
> >
> > I appreciate the clarification regarding the human evaluation protocol and model stability aspects. However, the concerns related to the main contributions of the paper have not been addressed.
> >
> > - The benchmark is the primary contribution; however, it appears somewhat limited. It mainly consists of 1,000 **culturally related prompts**. This raises concerns about its reusability for other researchers, since even human evaluators may make errors when assessing culturally specific knowledge with which they are unfamiliar. For example, “traditional cuisine of the Mid-Autumn Festival” may not be understood by many people, as not everyone is familiar with this festival. People may not even understand the solution without image references. Therefore, reference images are crucial for enabling accurate evaluation in such cases, even if they have some image variation.
> >
> > - Regarding the second main contribution, WiScore: it is largely based on the composition of scores provided by GPT-4. In my view, this composition does not introduce insights. Since models can easily become confused about objects, assigning higher weights to the consistency (i.e., object distinction) dimension will be a better alignment to ground truth.
> >
> > - Considering both points together, I am not fully convinced that the paper demonstrates sufficient novelty and rigor to warrant acceptance at a top machine learning venue.
> >
> > **Some additional suggestions for improving the paper and for future work:**
> >
> > In the related work section, more space could be devoted to text-to-image (T2I) evaluation, including existing evaluation methods and datasets given that the paper is about image generation evaluation. A more comprehensive inclusion of papers in this area would be beneficial. Currently, approximately two-thirds of the section is dedicated to introducing T2I models, which could be rebalanced.
> >
> > A more rigorous and transparent procedure for prompt selection can also be beneficial, for example, providing citations on the source for:
> >
> > > “We collected prompts from a variety of sources, including educational materials, encyclopedias, commonsense problem sets, and synthetic data generated by LLMs.”
> >
> > It would also be important to consider the coverage of cultural facts, such as the balance between mainstream and non-mainstream cultures, as well as representation across Western and Eastern contexts. A well-balanced dataset would help better identify potential cultural biases of LLMs.

---

> > > ### Author Response · Authors · 2026-04-03
> > >
> > > We thank the reviewer for the detailed follow-up and constructive suggestions.
> > >
> > > **1. Clarification on Benchmark Composition**
> > >
> > > We would like to respectfully clarify a critical detail regarding the dataset: WISE is **not** primarily composed of culturally related prompts. **Cultural Common Sense accounts for only 40% (400 prompts)**. The remaining **60% (600 prompts)** covers **Spatio-Temporal Reasoning (300)** and **Natural Science (300)**. These 600 prompts evaluate universally objective Spatio-Temporal and scientific laws and require absolutely no cultural familiarity to assess.
> > >
> > > **2. On the Necessity of Reference Images**
> > >
> > > To address concerns about unfamiliar cultures, we used **human evaluation WITH reference images** as the gold standard in the "Cultural Common Sense" category and compared other settings against it:
> > >
> > > | Evaluation Setting | Qwen-Image | UniWorld |
> > > | :--- | :--- | :--- |
> > > | **Baseline: Human eval WITH references** | **0.54** | **0.43** |
> > > | *Reference-based metric (CLIP similarity)* | 0.67 (+0.13) | 0.64 (+0.21) |
> > > | Human eval WITHOUT references | 0.52 (-0.02) | 0.41 (-0.02) |
> > > | **Our metric: WiScore** | **0.56 (+0.02)** | **0.43 (0.00)** |
> > >
> > > The data above reveals three clear conclusions:
> > > * **Image-based metrics are insufficient.** CLIP similarity deviates significantly from the human baseline. CLIP measures visual feature overlap but cannot reliably distinguish semantically different objects sharing similar visual traits. This validates the use of VLM-based semantic judgment over pixel-level comparison.
> > > * **Reference images do not meaningfully alter evaluation outcomes.** The scores from human evaluation without references deviate from the baseline by a maximum of 0.02. Model rankings remain identical. This proves that detailed textual Explanations and web search access fully resolve cultural familiarity issues for evaluators.
> > > * **WiScore highly aligns with the gold standard.** WiScore is nearly identical to human evaluation with reference images. This demonstrates that WiScore is a highly reliable and accurate metric for this task, successfully substituting the need for human annotators and explicit reference images.
> > >
> > > *(Note: We will add illustrative reference images to the appendix to aid general comprehension).*
> > >
> > > **3. Clarifying the Contribution of WiScore**
> > >
> > > We agree with the reviewer that the methodological formulation of WiScore is not a fundamentally new paradigm, and we will revise the paper to avoid positioning it as a standalone contribution. **The primary contribution of this paper is the WISE benchmark.** WiScore serves as a robust companion metric: it aligns well with human judgments (Krippendorff alpha: 0.82/0.78/0.67), preserves rankings across three VLM judges (GPT-4o, Gemini 3.1 Flash, Qwen3.5-122B), and is robust to weight variation (Spearman correlation $\ge 0.993$). We will clarify this role in the revision.
> > >
> > > **4. On Unique Positioning of WISE**
> > >
> > > WISE is the first benchmark to systematically evaluate **complex semantic understanding and world knowledge integration** in T2I generation. It differs fundamentally from prior work:
> > > * **T2I-CompBench** tests explicit constraints (e.g., "a red car next to a blue house"); the model simply renders the literal description.
> > > * **ScImage** evaluates structured scientific diagrams (e.g., "a 6-by-3 matrix") and code-based generation (Python/TikZ), a different paradigm from natural image generation.
> > > * **WISE** is a benchmark for world knowledge-informed semantic evaluation, assessing whether text-to-image models can integrate complex semantic understanding and world knowledge across 25 subdomains.
> > >
> > > **5. Implementation of Additional Suggestions**
> > >
> > > We fully accept the excellent suggestions from the reviewer to improve the manuscript:
> > > * **Related Work:** We will significantly expand the discussion on T2I evaluation methods (including MMMG and WorldGenBench) and condense the model introduction section.
> > > * **Prompt Sourcing:** We will provide an explicit table citing the sources for all prompts (educational materials, encyclopedias, etc.) and detail the quality control procedures.
> > > * **Cultural Balance Analysis:** We have mapped the cultural distribution of all 1,000 prompts. **56.6% are Global/Neutral**, **22.6% are Western**, and **20.8% are Non-Western**. This demonstrates that WISE maintains a healthy balance and does not exhibit a strong Western or Eastern skew. We will include this analysis to help identify cultural biases in LLMs.
> > >
> > > We believe these revisions and new experiments address the remaining concerns, and we hope the reviewer will reconsider the novelty and rigor of our contribution.
> > >
> > > **Anonymous Resources:** To facilitate verification of our supplementary experiments (alternative VLM judge code, human evaluation guidelines, and human evaluation page with reference images), we have uploaded the materials to this anonymous repository: `https://anonymous.4open.science/r/ICML_rebuttal_20159-430E`

---

### Official Review · Reviewer_AdvP · 2026-03-12

**Soundness:** 1
**Presentation:** 3
**Significance:** 2
**Originality:** 2
**Overall Recommendation:** 3
**Confidence:** 4

**Summary:**

The paper introduces WISE, a benchmark of 1000 prompts testing whether T2I models can integrate world knowledge into their generations. Prompts span 25 subdomains across cultural common sense, spatiotemporal reasoning, and natural science. WiScore, a composite metric using GPT as judge, rates images on consistency, realism, and aesthetic quality. Twenty models are evaluated (10 dedicated T2I, 10 unified multimodal). Most score below 0.6. When prompts are rewritten by GPT-4o into simpler, explicit forms (e.g., "the plant often gifted on Mother's Day" becomes "Carnation"), scores improve substantially across the board, with BAGEL on rewritten prompts (0.73) nearly matching BAGEL+CoT on originals (0.70).

**Compliance With Llm Reviewing Policy:**

Affirmed.

**Final Justification:**

The rebuttal resolved my top concern: rankings are stable across three VLM judges and weight configurations (Spearman ≥ 0.993). Resolution confound and human evaluation protocol are also adequately addressed.

The core issue (Q2) remains open. Table 2 shows mean Δ = +0.23 after rewriting, with weaker models gaining most. One anecdotal rewrite-resistant example does not establish what fraction of WISE measures world knowledge vs. prompt comprehension. This distinction is central to the paper's claim. No statistical analysis (seed variance, confidence intervals) is provided for 40-prompt subdomains, and cultural bias interactions between prompt set and judge are undiscussed.

Score raised from 2 to 3 reflecting resolved robustness concerns. Cannot go higher without empirical separation of knowledge and comprehension effects.

**Key Questions For Authors:**

1. Please report Spearman rank correlation of model rankings under three weight configurations: (0.7/0.2/0.1), (0.5/0.3/0.2), (0.4/0.4/0.2). Also rerun WiScore with at least one alternative VLM judge (e.g., Gemini 1.5 Pro or an open-source VLM). If rankings change, the conclusions depend on arbitrary choices. This is the most important question for my assessment.

2. Table 2 shows the main bottleneck is prompt comprehension, not knowledge. How do you reconcile this with the "world knowledge" framing? Can you design prompts where no simple rewrite is possible? Showing that a rewrite-resistant subset yields different conclusions would change my evaluation.

3. Please report: (a) number of human annotators, (b) inter-annotator agreement (Krippendorff's alpha), (c) annotator instructions, (d) whether any annotators had natural science domain expertise.

4. How do you account for the resolution confound? Models at 1024x1024 will score higher on realism and aesthetics than those at 512x512 regardless of knowledge.

**Limitations:**

No. The authors note models may reflect training data biases. They do not discuss the reproducibility risk of a closed-source judge, cultural bias in the prompt set, the prompt comprehension vs. knowledge confound from their own rewriting experiment, lack of statistical testing, data contamination risk, the resolution confound, or concurrent work addressing the same gap.

**Strengths And Weaknesses:**

Strengths:

- WISE targets a real gap. Existing benchmarks (GenEval, T2I-CompBench) test compositional ability with explicit prompts. WISE requires resolving implicit references ("the signature instrument of the rock and roll era in the 1950s") or reasoning about unstated facts ("a candle in space").
- The rewritten prompts experiment (Table 2) is the paper's best finding. It cleanly shows that much of the apparent "knowledge gap" is a prompt comprehension gap. BAGEL+CoT on originals (0.70) nearly matches BAGEL without CoT on rewritten prompts (0.73), showing CoT acts as an internal prompt simplifier. This is actionable for model developers.
- Broad model coverage: 20 models with useful architectural categorization (autoregressive, cascaded AR+diffusion, parallel AR+diffusion).
- The paper is clearly written and Figure 3 illustrates the taxonomy well.

Weaknesses:

- WiScore relies on a single closed-source, versioned API model. GPT-4o-2024-05-13 will be deprecated. No alternative judge is tested. Other researchers cannot reproduce these scores.
- The metric weights (0.7/0.2/0.1) have no justification. No sensitivity analysis is provided. If rankings shift under different weights, the conclusions are fragile.
- Table 2 undermines the paper's framing. If a model generates a correct carnation from "carnation" but not from "the plant often gifted on Mother's Day," it has the knowledge but cannot parse the words where fewer words would do. WISE then measures prompt comprehension, not world knowledge. The paper does not resolve this tension.
- Improvement magnitudes in Table 2 correlate inversely with baseline performance (Janus-Pro-1B: +0.34 vs BAGEL: +0.21). Weaker models are penalized by prompt complexity, not knowledge deficits.
- Cultural Common Sense accounts for 40% of prompts. The examples show East Asian and Western bias ("Traditional food of the Mid-Autumn Festival," "A famous flower that symbolizes wealth in China," "The iconic hat of the protagonist of One Piece"). GPT-4o has documented accuracy disparities of up to 58% across cultural settings. A culturally biased judge evaluating culturally biased prompts compounds the problem.
- Models are compared at their default resolutions (512x512 to 1024x1024). Higher-resolution images systematically score better on realism and aesthetics, introducing a confound unrelated to knowledge.
- Human evaluation is undocumented: annotator count, agreement rates, instructions, and domain expertise are unreported. Validation on 100 images from one model is insufficient.
- 40 prompts per subdomain is too few for per-subdomain conclusions given T2I stochasticity. No confidence intervals or seed variance reported.
- PhyBench (Meng et al., 2024) tests physical commonsense, Commonsense-T2I (Fu et al., 2024) tests common sense. MMMG (NeurIPS 2025, 4456 prompts) and WorldGenBench address the same gap and are not cited.
- No methodological contribution beyond benchmark and metric design.

---

> ### Author Rebuttal · Authors · 2026-03-31
>
> We sincerely thank you for the insightful and constructive suggestions! Due to space limitations, we have addressed the critical questions here; however, we remain fully open to providing further clarifications on any other points during the discussion phase.
>
> **Q1 & W1/W2**: Stability of WiScore Using Different Models and Different Weights
>
> **A:** Thank you for this important suggestion. We use Gemini 3.1 Flash and the open-source Qwen3.5-122B-A10B as alternative judges. Due to rebuttal time and compute resources limits, we used representative models from the Qwen-Image, UniWorld, FLUX, Stable Diffusion, and Janus families. The overall ranking is stable across judges:
>
> | Model | Original | Gemini | Qwen |
> |---|---|---|---|
> | Qwen-Image | #1 (0.62) | #1 (0.50) | #1 (0.57) |
> | UniWorld-V1 | #2 (0.55) | #2 (0.44) | #2 (0.45) |
> | FLUX.1-dev | #3 (0.50) | #3 (0.43) | #2 (0.45) |
> | SD-3.5-large | #4 (0.46) | #4 (0.40) | #4 (0.44) |
> | SD-3.5-medium | #5 (0.45) | #5 (0.37) | #5 (0.42) |
> | SD-XL-base-0.9 | #6 (0.43) | #6 (0.36) | #6 (0.40) |
> | SD-3-medium | #7 (0.42) | #6 (0.36) | #6 (0.40) |
> | FLUX.1-schnell | #8 (0.40) | #8 (0.34) | #9 (0.37) |
> | Janus-Pro-7B | #9 (0.35) | #9 (0.30) | #8 (0.38) |
> | SD-v1-5 | #10 (0.32) | #10 (0.28) | #10 (0.34) |
> | SD-2-1 | #10 (0.32) | #11 (0.27) | #10 (0.34) |
> | Janus-Pro-1B | #12 (0.26) | #12 (0.23) | #13 (0.28) |
> | Janus-1.3B | #13 (0.23) | #13 (0.20) | #12 (0.29) |
>
>
> We also tested alternative weights (0.5/0.3/0.2) and (0.4/0.4/0.2). The Spearman rank correlations with the original (0.7/0.2/0.1) are both 0.993, with only one adjacent swap (SD-3-medium vs. SD-XL-base-0.9), confirming that our conclusions are robust to both judge and weight choices. We will include these results in the revision and release the alternative evaluation pipeline.
>
> **Q2 & W3**: Table 2 Suggests the Benchmark Measures Prompt Comprehension Rather Than World Knowledge
>
> **A:** Thank you for this profound question. We consider the factual association that “carnation is the representative plant of Mother’s Day” as world knowledge itself, rather than mere language comprehension. The improvement in Table 2 after prompt rewriting arises because rewriting converts the original task from implicit knowledge inference (Mother’s Day plant → carnation) into an explicit entity-generation task, thereby substantially lowering the difficulty. Therefore, Table 2 does not weaken our framing; instead, it shows that current models struggle to invoke and apply knowledge under implicit, knowledge-intensive prompts.
> Regarding rewrite-resistant prompts, some cases remain knowledge-dependent even after rewriting. For example, for “An animal cell in late anaphase of mitosis, showing the movement of its chromosomes,” even a rewritten version such as “A cell where sister chromatids have separated and are being pulled toward opposite poles by spindle fibers” still requires the model to understand the spatial relationships among spindle fibers, chromosomes, and cell poles. Without such biological knowledge, the generated image is likely to remain structurally incorrect. We will clarify this definition of “world knowledge” and further analyze such rewrite-resistant cases in the revision.
>
> **Q3 & W6**: Human Evaluation Protocol
>
> **A:** Thank you for this important suggestion. We provide the following details: each image is evaluated by 5 independent annotators; all annotators score each image along the three dimensions of consistency, realism, and aesthetics; the final score is computed using the same weighting scheme as WiScore; and each sample's result is obtained by averaging all annotators' scores. Regarding annotator background, we will clarify in the revision: our annotators are at the undergraduate level or phd level. Although we did not screen annotators by specific disciplinary expertise, they were allowed to use search engines for verification.
> We have also expanded the human evaluation to 180 images each from Qwen-Image and UniWorld, reducing the bias caused by single-model, small-sample validation. The following results are reported on the 180-image evaluation subset:
> | Model | WiScore | Human Score |
> |---|---|---|
> | Qwen-Image | 0.50 | 0.54 |
> | UniWorld  | 0.40 | 0.42 |
>
> | Dimension | Krippendorff's α |
> |---|---|
> | Consistency | 0.82 |
> | Realism | 0.78 |
> | Aesthetics | 0.67 |
>
> The high inter-annotator agreement and close alignment between human and WiScore values demonstrate the reliability of our evaluation.
>
> **Q4 & W5**: Resolution Confound
>
> **A:** We reevaluated representative models (Qwen-Image, FLUX, SD series) at a unified 512 resolution. The ranking remains unchanged except SD-3.5-medium surpasses SD-3.5-large by 0.01, confirming resolution is not the main driver of our ranking conclusions on WISE.
>
> **W8**: Missing Related Work
>
> **A:** Thank you for the reminder. We actually already discuss PhyBench and Commonsense-T2I in related work, and we promise to add MMMG and WorldGenBench in the revised version.

---

> > ### Author Rebuttal · Reviewer_AdvP · 2026-04-04
> >
> > The judge stability results mostly resolve my biggest concern: rankings are consistent across different VLMs, and robust to weight variation. The resolution confound is adequately addressed, and the expanded human evaluation with reported Krippendorff's α values is a meaningful improvement.
> >
> > However, I have a follow-up on Q2. The authors argue that associating "carnation" with "Mother's Day" constitutes world knowledge, and I partially agree. But the near-universal improvement after rewriting raises a question the rebuttal does not answer: what fraction of WISE prompts are genuinely knowledge-bottlenecked (small Δ after rewriting) vs. comprehension-bottlenecked (large Δ)? The mitosis example is illustrative but anecdotal. Could the authors partition WISE by per-prompt rewriting gain and report whether model rankings or relative strengths differ across these two subsets? This would clarify whether WISE primarily measures indirect reference resolution or world knowledge integration, which is the core claim of the paper.
> >
> > Additionally, the authors noted they were open to addressing further points during discussion. I would appreciate brief responses to the following from my original review that were not covered in the rebuttal: confidence intervals or seed variance given 40 prompts per subdomain, and how the authors account for documented cross-cultural accuracy disparities in GPT-4o when it serves as judge over a prompt set where 40% is cultural common sense.
> >
> > I am raising my score from 2 to 3 to reflect the resolved robustness concerns.

---

> > > ### Author Response · Authors · 2026-04-07
> > >
> > > We sincerely thank the reviewer for the detailed follow-up and for the constructive suggestions.
> > >
> > > ---
> > >
> > > ### 1. Follow-up on Q2: rewrite gain analysis
> > >
> > > Following the reviewer’s suggestion, for each prompt, we compute its **across-model mean rewriting gain**, i.e., the average score improvement from the original prompt to the rewritten prompt across models, and use this value to partition prompts into **high-gain** and **low-gain** subsets. Their Spearman correlations with the overall ranking are **0.9455** and **0.9818**, respectively, indicating that both subsets remain highly consistent with the overall ranking.
> > >
> > > However, gain alone can be misleading: prompts with already high original scores may show small gains and be incorrectly treated as rewrite-ineffective, even though rewriting is unnecessary.
> > >
> > > We further use a **threshold-based partition**:
> > > * **(1) rewrite-effective (Eff):** original < $T$, rewritten $\ge T$;
> > > * **(2) rewrite-ineffective (Ineff):** original < $T$, rewritten < $T$;
> > > * **(3) not-low-original:** original $\ge T$.
> > >
> > > Using $T=0.6$, among originally difficult prompts, **39.9%** are rewrite-effective while **60.1%** remain rewrite-ineffective. This means rewriting helps, but does **not** resolve the majority of difficult WISE prompts.
> > >
> > > The pattern is category-dependent: at $T=0.6$, **56.8%** of difficult **Cultural Common Sense** prompts are resolved by rewriting, but only **14.9%** of **Physics** prompts are resolved.
> > >
> > > | Category | Effective Ratio | Ineffective Ratio |
> > > | :--- | :---: | :---: |
> > > | Cultural Common Sense | 56.8% | 43.2% |
> > > | Time | 37.6% | 62.4% |
> > > | Space | 30.8% | 69.2% |
> > > | Biology | 28.9% | 71.1% |
> > > | Physics | 14.9% | 85.1% |
> > > | Chemistry | 26.0% | 74.0% |
> > >
> > > We further verify the **overall conclusion** across multiple thresholds $T$: the rewrite-ineffective subset consistently remains larger than the rewrite-effective subset, and its ranking stays highly consistent with the overall ranking.
> > >
> > > | Threshold $T$ | Ratio (Eff / Ineff) | Spearman (Eff / Ineff) |
> > > | :---: | :---: | :---: |
> > > | **0.6** | 34.3% / 51.6% | 0.9273 / 1.0000 |
> > > | **0.7** | 24.8% / 66.7% | 0.9636 / 0.9909 |
> > > | **0.8** | 13.6% / 83.2% | 0.9182 / 0.9909 |
> > >
> > > Overall, WISE does include prompts that benefit from rewriting, but also contains a **rewrite-ineffective subset** that remains difficult after rewriting. Thus, WISE is **not primarily driven by rewrite-removable effects**, but also captures persistent **knowledge-informed generation** difficulty.
> > >
> > > ---
> > >
> > > ### 2. Confidence intervals / seed variance
> > >
> > > We would also like to clarify that our main conclusions are reported at the six-category and overall levels, rather than relying on fine-grained ranking claims over all 25 subdomains; correspondingly, the paper’s core claims are not based on “40 prompts per subdomain” comparisons.
> > >
> > > To address stochasticity directly, we ran **5-seed evaluation** (42–46) and computed mean/std and 95% confidence intervals. The overall ranking is highly stable across seeds.
> > >
> > > | Model | Mean ± Std | 95% CI | Rank range |
> > > | :--- | :--- | :--- | :---: |
> > > | Qwenimage | 0.5029 ± 0.0046 | [0.4972, 0.5086] | 1-1 |
> > > | FLUX.1-dev | 0.4225 ± 0.0045 | [0.4168, 0.4281] | 2-2 |
> > > | SD-3.5-large | 0.4040 ± 0.0092 | [0.3926, 0.4154] | 3-3 |
> > > | SD-3.5-medium | 0.3714 ± 0.0015 | [0.3696, 0.3733] | 4-4 |
> > > | SD-3-medium | 0.3584 ± 0.0042 | [0.3532, 0.3636] | 5-6 |
> > > | SD-XL-0.9 | 0.3584 ± 0.0087 | [0.3476, 0.3692] | 5-7 |
> > > | FLUX.1-schnell | 0.3388 ± 0.0231 | [0.3102, 0.3675] | 5-7 |
> > > | Janus-Pro-7B | 0.3043 ± 0.0014 | [0.3026, 0.3060] | 8-8 |
> > > | SD-1.5 | 0.2758 ± 0.0084 | [0.2654, 0.2862] | 9-9 |
> > > | Janus-Pro-1B | 0.2339 ± 0.0028 | [0.2305, 0.2374] | 10-10 |
> > > | Janus-1.3B | 0.2080 ± 0.0054 | [0.2013, 0.2147] | 11-11 |
> > >
> > > Thus, stochasticity does **not** affect the principal comparative conclusions.
> > >
> > > ---
> > >
> > > ### 3. Cross-cultural validity of GPT-4o as judge
> > >
> > > We audited the **400 Cultural Common Sense prompts** in WISE: **41.50% Western, 40.75% non-Western, and 17.75% global/neutral**, indicating that this subset is balanced rather than dominated by a single cultural bloc.
> > >
> > > To examine GPT-4o as a judge on culturally informed prompts, we constructed a **160-image human-evaluation subset** from the Cultural Common Sense category.
> > >
> > > | Evaluation Setting | Qwen-Image | UniWorld |
> > > | :--- | :---: | :---: |
> > > | Human eval | 0.52 | 0.41 |
> > > | WiScore | 0.56 | 0.43 |
> > >
> > > WiScore remains close to human evaluation on this subset, suggesting that the **explanations** paired with WISE prompts reduce ambiguity and limit dependence on prior cultural familiarity. In addition, the rankings remain stable when GPT-4o is replaced by **Gemini 3.1 Flash** and **Qwen3.5-122B-A10B**.
> > >
> > > We will release the **Gemini/Qwen evaluation implementations** and **update the leaderboard accordingly** to improve reproducibility.
> > >
> > > We believe these additional analyses and experiments address the reviewer’s follow-up concerns, and we sincerely thank the reviewer again for helping us substantially strengthen the paper.

---

### Official Review · Reviewer_WYXs · 2026-03-13

**Soundness:** 3
**Presentation:** 3
**Significance:** 4
**Originality:** 3
**Overall Recommendation:** 4
**Confidence:** 4

**Summary:**

This paper introduces WISE (World Knowledge-Informed Semantic Evaluation), a novel benchmark designed to evaluate text-to-image (T2I) models' ability to integrate and apply world knowledge during image generation. The benchmark contains 1,000 carefully crafted prompts across 25 subdomains in three major areas: cultural common sense, spatio-temporal reasoning, and natural science. The authors also propose WiScore, a new composite metric that evaluates consistency, realism, and aesthetic quality. Through comprehensive evaluation of 20 models (10 dedicated T2I models and 10 unified multimodal models), the paper reveals significant limitations in current models' world knowledge integration capabilities.

**Compliance With Llm Reviewing Policy:**

Affirmed.

**Ethical Review Concerns:**

My concerns have been adequately addressed.

**Final Justification:**

My concerns have been adequately addressed.

**Key Questions For Authors:**

1. The paper shows that unified multimodal models with strong understanding capabilities don't necessarily perform better on WISE. Have you investigated whether fine-tuning these models specifically on knowledge-intensive tasks could bridge this gap? This would help determine if the issue is architectural or simply a matter of training data distribution.

2. The WiScore metric weights consistency at 0.7, which seems reasonable for knowledge evaluation. However, did you experiment with different weight configurations? How sensitive are the overall rankings to these weight choices?

3. The benchmark currently focuses on static image generation. Have you considered extending WISE to evaluate multi-image generation (e.g., for temporal sequences) or interactive generation scenarios?

**Limitations:**

The authors adequately discuss limitations including: (1) potential category overlap in knowledge domains, (2) the evolving nature of world knowledge, and (3) some models not being evaluated due to API availability. The limitations discussion is honest and constructive.

**Strengths And Weaknesses:**

The paper is technically sound with a well-designed evaluation methodology.

---

> ### Author Rebuttal · Authors · 2026-03-31
>
> **Q1**: Targeted Fine-tuning for Knowledge-intensive Tasks
>
> **A:** Thank you for your suggestion. To investigate this question, we conducted a supplementary experiment: we constructed 115K synthetic samples targeting knowledge- and reasoning-intensive generation tasks, and fine-tuned the unified multimodal model BAGEL accordingly. The data was automatically generated by a large language model using a triplet structure of (implicit prompt, reasoning chain, explicit prompt), covering knowledge-intensive scenarios (e.g., physics, chemistry, biology, spatiotemporal relations, entity knowledge) as well as multi-step reasoning tasks (e.g., arithmetic reasoning, constraint satisfaction, causal reasoning), with explicit intermediate reasoning processes provided as supervision signals.
>
> | Benchmark | Before Fine-tuning | After Fine-tuning | Δ |
> |---|---|---|---|
> | WISE | 0.52 | 0.56 | +0.04 |
> | R2I-Bench | 0.36 | 0.44 | +0.08 |
>
> The results show that targeted fine-tuning can improve performance to some extent, but does not fully resolve the challenges of knowledge-driven generation. We believe this issue is likely a joint effect of architectural capacity and data/training paradigm: on one hand, data and supervision formats are critical for activating a model's knowledge and reasoning capabilities; on the other hand, existing architectures and training paradigms still have limitations in effectively transferring understanding capabilities into generation capabilities.
>
> **Q2**: Stability of WiScore Using Different Weights
>
> **A:** Thank you for this important suggestion. We tested alternative weights, changing (0.7/0.2/0.1) to (0.5/0.3/0.2) and (0.4/0.4/0.2).
> | Model | (0.7/0.2/0.1) | (0.5/0.3/0.2) | (0.4/0.4/0.2) |
> | :--- | :--- | :--- | :--- |
> | Qwenimage | #1 (0.50) | #1 (0.52) | #1 (0.53) |
> | UniWorld-V1 | #2 (0.44) | #2 (0.46) | #2 (0.46) |
> | FLUX.1-dev | #3 (0.43) | #3 (0.45) | #2 (0.46) |
> | stable-diffusion-3.5-large | #4 (0.40) | #4 (0.41) | #4 (0.42) |
> | stable-diffusion-3.5-medium | #5 (0.37) | #5 (0.39) | #5 (0.40) |
> | stable-diffusion-3-medium-diffusers | #6 (0.36) | #7 (0.37) | #7 (0.37) |
> | stable-diffusion-xl-base-0.9 | #7 (0.36) | #6 (0.38) | #6 (0.38) |
> | FLUX.1-schnell | #8 (0.34) | #8 (0.35) | #8 (0.36) |
> | Janus-Pro-7B | #9 (0.30) | #9 (0.31) | #9 (0.31) |
> | stable-diffusion-v1-5 | #10 (0.28) | #10 (0.28) | #10 (0.27) |
> | stable-diffusion-2-1 | #11 (0.27) | #11 (0.28) | #11 (0.27) |
> | Janus-Pro-1B | #12 (0.23) | #12 (0.23) | #12 (0.22) |
> | Janus-1.3B | #13 (0.20) | #13 (0.19) | #13 (0.18) |
>
> The Spearman rank correlations are extremely high: 0.993 between the original weighting and each alternative, and 1.000 between the two alternatives, indicating that the model rankings are nearly unchanged across different weight choices. Except for one adjacent swap between SD-3-medium and SD-XL-base-0.9, all other ranks remain unchanged. These results show that evaluation results are robust to both judge choice and weight choice. We will include these results in the revision.
>
>
> **Q3**: Multi-image Extension of the Benchmark
>
> **A:** Thank you for your suggestion. We agree that extending WISE to multi-image generation (e.g., temporal sequences) or interactive generation scenarios is an important and promising direction. We believe the overall framework of WISE has good extensibility, particularly toward the following directions:
>
> (1) Multi-image generation grounded in cultural/historical knowledge: for example, requiring a model to generate a set of images depicting changes in clothing or lifestyle across different historical periods (e.g., the evolution of French men's fashion from the late 19th century to the mid-20th century to the present), and evaluating knowledge consistency and temporal plausibility across images.
>
> (2) Temporal generation of scientific processes: for example, generating a sequence of images describing a biological or physical process (e.g., the morphological transformation from caterpillar to butterfly), thereby evaluating causal consistency and scientific plausibility across a temporal sequence.
>
> We believe these directions require additional benchmark design (e.g., sequence-level annotation, cross-image evaluation metrics) and represent natural extensions of the current work, but are beyond the scope of this paper. We will add a discussion of these directions in the revision and clarify that the current benchmark, as a first step, primarily focuses on static image generation scenarios.

---

> > ### Author Rebuttal · Reviewer_WYXs · 2026-04-02
> >
> > My concerns have been adequately addressed.

---

> > > ### Author Response · Authors · 2026-04-02
> > >
> > > Thank you very much for your feedback and for acknowledging that your concerns have been fully resolved.
> > >
> > > Given that the concerns have been fully addressed, we kindly wonder if you would consider increasing the score to better reflect the current state of the paper. We believe that your insightful suggestions have helped significantly strengthen the work and its potential impact on the community.
> > >
> > > In any case, we sincerely thank you again for your careful review and constructive feedback.

---

### Decision · Program_Chairs · 2026-04-30

**Decision:**

Accept (regular)

**Comment:**

This paper proposes WISE, a new benchmark for evaluating the integration of World Knowledge in text-to-image (T2I) models, alongside its evaluation metric, WiScore. While traditional benchmarks focus primarily on faithfulness to explicit instructions, WISE consists of 1,000 prompts that require implicit reasoning and cultural/scientific knowledge. A significant contribution is that through extensive experiments on 20 state-of-the-art models, the study highlights the limitations of current T2I models in knowledge utilization.
Although some reviewers remain cautious regarding the complete separation of knowledge and understanding, the practicality as an engineering benchmark and the depth of analysis sufficiently meet the acceptance criteria of top-tier conferences.

In the camera-ready version, the authors are required to integrate the following elements presented in the rebuttal into the main text and appendix such as the detailed analysis of prompt rewriting, stability across multiple seeds, and details of the human evaluation protocol. Furthermore, the refinement of model architecture definitions, as pointed out by reviewer QvBt, should also be reflected.

This paper fills a critical gap in the evaluation of generative AI. Given that the dataset and metrics are likely to be widely utilized in future T2I research, the AC recommends acceptance.